# HyperDeepONet: learning operator with complex target function space using the limited resources via hypernetwork

**Jae Yong Lee**[*]
Center for Artificial Intelligence and Natural Sciences
Korea Institute for Advanced Study
Seoul, 02455, Republic of Korea
{jaeyong}@kias.re.kr

**Sung Woong Cho**[*]**& Hyung Ju Hwang** [†]
Department of Mathematics
Pohang University of Science and Technology
Pohang, 37673, Republic of Korea
{swcho95kr,hjhwang}@postech.ac.kr

## Abstract

Fast and accurate predictions for complex physical dynamics are a significant challenge across various applications. Real-time prediction on resource-constrained hardware is even more crucial in real-world problems. The deep operator network (DeepONet) has recently been proposed as a framework for learning nonlinear mappings between function spaces. However, the DeepONet requires many parameters and has a high computational cost when learning operators, particularly those with complex (discontinuous or non-smooth) target functions. This study proposes HyperDeepONet, which uses the expressive power of the hypernetwork to enable the learning of a complex operator with a smaller set of parameters. The DeepONet and its variant models can be thought of as a method of injecting the input function information into the target function. From this perspective, these models can be viewed as a particular case of HyperDeepONet. We analyze the complexity of DeepONet and conclude that HyperDeepONet needs relatively lower complexity to obtain the desired accuracy for operator learning. HyperDeepONet successfully learned various operators with fewer computational resources compared to other benchmarks.

## 1 Introduction

Operator learning for mapping between infinite-dimensional function spaces is a challenging problem. It has been used in many applications, such as climate prediction (Kurth et al., 2022) and fluid dynamics (Guo et al., 2016). The computational efficiency of learning the mapping is still important in real-world problems. The target function of the operator can be discontinuous or sharp for complicated dynamic systems. In this case, balancing model complexity and cost for computational time is a core problem for the real-time prediction on resource-constrained hardware (Choudhary et al., 2020; Murshed et al., 2021).

Many machine learning methods and deep learning-based architectures have been successfully developed to learn a nonlinear mapping from an infinite-dimensional Banach space to another. They focus on learning the solution operator of some partial differential equations (PDEs), e.g., the initial or boundary condition of PDE to the corresponding solution. Anandkumar et al. (2019) proposed an iterative neural operator scheme to learn the solution operator of PDEs.

Simultaneously, Lu et al. (2019; 2021) proposed a deep operator network (DeepONet) architecture based on the universal operator approximation theorem of Chen & Chen (1995). The DeepONet consists of two networks: branch net taking an input function at fixed finite locations, and trunk net taking a query location of the output function domain. Each network provides the $p$ outputs. The two $p$-outputs are combined as a linear combination (inner-product) to approximate the underlying operator, where the branch net produces the coefficients ($p$-coefficients) and the trunk net produces the basis functions ($p$-basis) of the target function.

---

[*]These authors contributed equally.
[†]corresponding author

While variant models of DeepONet have been developed to improve the vanilla DeepONet, they still have difficulty approximating the operator for a complicated target function with limited computational resources. Lanthaler et al. (2022) and Kovachki et al. (2021b) pointed out the limitation of linear approximation in DeepONet. Some operators have a slow spectral decay rate of the Kolmogorov $n$-width, which defines the error of the best possible linear approximation using an $n-$dimensional space. A large $n$ is required to learn such operators accurately, which implies that the DeepONet requires a large number of basis $p$ and network parameters for them.

Hadorn (2022) investigated the behavior of DeepONet, to find what makes it challenging to detect the sharp features in the target function when the number of basis $p$ is small. They proposed a Shift-DeepONet by adding two neural networks to shift and scale the input function. Venturi & Casey (2023) also analyzed the limitation of DeepONet via singular value decomposition (SVD) and proposed a flexible DeepONet (flexDeepONet), adding a pre-net and an additional output in the branch net. Recently, to overcome the limitation of the linear approximation, Seidman et al. (2022) proposed a nonlinear manifold decoder (NOMAD) framework by using a neural network that takes the output of the branch net as the input along with the query location. Even though these methods reduce the number of basis functions, the total number of parameters in the model cannot be decreased. The trunk net still requires many parameters to learn the complex operators, especially with the complicated (discontinuous or non-smooth) target functions.

In this study, we propose a new architecture, HyperDeepONet, which enables operator learning, and involves a complex target function space even with limited resources. The HyperDeepONet uses a hypernetwork, as proposed by Ha et al. (2017), which produces parameters for the target network. Wang et al. (2022) pointed out that the final inner product in DeepONet may be inefficient if the information of the input function fails to propagate through a branch net. The hypernetwork in HyperDeepONet transmits the information of the input function to each target network's parameters. Furthermore, the expressivity of the hypernetwork reduces the neural network complexity by sharing the parameters (Galanti & Wolf, 2020). Our main contributions are as follows.

- We propose a novel HyperDeepONet using a hypernetwork to overcome the limitations of DeepONet and learn the operators with a complicated target function space. The DeepONet and its variant models are analyzed primarily in terms of expressing the target function as a neural network (Figure 4). These models can be simplified versions of our general HyperDeepONet model (Figure 5).

- We analyze the complexity of DeepONet (Theorem 2) and prove that the complexity of the HyperDeepONet is lower than that of the DeepONet. We have identified that the DeepONet should employ a large number of basis to obtain the desired accuracy, so it requires numerous parameters. For variants of DeepONet combined with nonlinear reconstructors, we also present a lower bound for the number of parameters in the target network.

- The experiments show that the HyperDeepONet facilitates learning an operator with a small number of parameters in the target network even when the target function space is complicated with discontinuity and sharpness, which the DeepONet and its variants suffer from. The HyperDeepONet learns the operator more accurately even when the total number of parameters in the overall model is the same.

## 2 RELATED WORK

Many machine learning methods and deep learning-based architectures have been successfully developed to solve PDEs with several advantages. One research direction is to use the neural network directly to represent the solution of PDE (E & Yu, 2018; Sirignano & Spiliopoulos, 2018). The physics-informed neural network (PINN), introduced by Raissi et al. (2019), minimized the residual of PDEs by using automatic differentiation instead of numerical approximations.

There is another approach to solve PDEs, called operator learning. Operator learning aims to learn a nonlinear mapping from an infinite-dimensional Banach space to another. Many studies utilize the convolutional neural network to parameterize the solution operator of PDEs in various applications (Guo et al., 2016; Bhatnagar et al., 2019; Khoo et al., 2021; Zhu et al., 2019; Hwang et al., 2021). The neural operator (Kovachki et al., 2021b) is proposed to approximate the nonlinear operator inspired by Green's function. Li et al. (2021) extend the neural operator structure to the Fourier Neural

Operator (FNO) to approximate the integral operator effectively using the fast Fourier transform (FFT). Kovachki et al. (2021a) proved the universality of FNO and identified the size of the network.

The DeepONet (Lu et al., 2019; 2021) has also been proposed as another framework for operator learning. The DeepONet has significantly been applied to various problems, such as bubble growth dynamics (Lin et al., 2021), hypersonic flows (Mao et al., 2021), and fluid flow (Cai et al., 2021).

Lanthaler et al. (2022) provided the universal approximation property of DeepONet. Wang et al. (2021) proposed physics-informed DeepONet by adding a residual of PDE as a loss function, and Ryck & Mishra (2022) demonstrated the generic bounds on the approximation error for it. Prasthofer et al. (2022) considered the case where the discretization grid of the input function in DeepONet changes by employing the coordinate encoder. Lu et al. (2022) compared the FNO with DeepONet in different benchmarks to demonstrate the relative performance. FNO can only infer the output function of an operator as the input function in the same grid as it needs to discretize the output function to use Fast Fourier Transform(FFT). In contrast, the DeepONet can predict from any location.

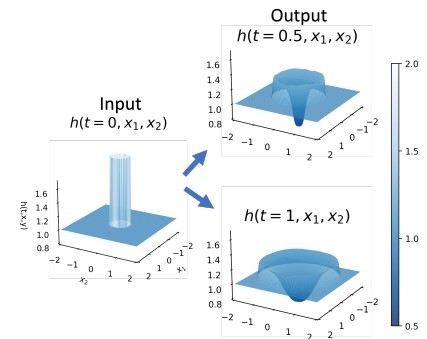

Figure 1: Example of operator learning: the input function and the output function for the solution operator of shallow water equation

Ha et al. (2017) first proposed hypernetwork, a network that creates a weight of the primary network. Because the hypernetwork can achieve weight sharing and model compression, it requires a relatively small number of parameters even as the dataset grows. Galanti & Wolf (2020) proved that a hypernetwork provides higher expressivity with low-complexity target networks. Sitzmann et al. (2020) and Klocek et al. (2019) employed this approach to restoring images with insufficient pixel observations or resolutions. de Avila Belbute-Peres et al. (2021) investigated the relationship between the coefficients of PDEs and the corresponding solutions. They combined the hypernetwork with the PINN's residual loss. For time-dependent PDE, Pan et al. (2022) designated the time $t$ as the input of the hypernetwork so that the target network indicates the solution at $t$. von Oswald et al. (2020) devised a chunk embedding method that partitions the parameters of the target network; this is because the output dimension of the hypernetwork can be large.

## 3 OPERATOR LEARNING

### 3.1 PROBLEM SETTING

The goal of operator learning is to learn a mapping from infinite-dimensional function space to the others using a finite pair of functions. Let $\mathcal{G} : \mathcal{U} \to \mathcal{S}$ be a nonlinear operator, where $\mathcal{U}$ and $\mathcal{S}$ are compact subsets of infinite-dimensional function spaces $\mathcal{U} \subset C(\mathcal{X}; \mathbb{R}^{d_u})$ and $\mathcal{S} \subset C(\mathcal{Y}; \mathbb{R}^{d_s})$ with compact domains $\mathcal{X} \subset \mathbb{R}^{d_x}$ and $\mathcal{Y} \subset \mathbb{R}^{d_y}$. For simplicity, we focus on the case $d_u = d_s = 1$, and all the results could be extended to a more general case for arbitrary $d_u$ and $d_s$. Suppose we have observations $\{u_i, \mathcal{G}(u_i)\}_{i=1}^N$ where $u_i \in \mathcal{U}$ and $\mathcal{G}(u_i) \in \mathcal{S}$. We aim to find an approximation $\mathcal{G}_\theta : \mathcal{U} \to \mathcal{S}$ with parameter $\theta$ using the $N$ observations so that $\mathcal{G}_\theta \approx \mathcal{G}$. For example, in a dam break scenario, it is an important to predict the fluid flow over time according to a random initial height of the fluid.

To this end, we want to find an operator $\mathcal{G}_\theta$, which takes an initial fluid height as an input function and produces the fluid height over time at any location as the output function (Figure 1).

As explained in Lanthaler et al. (2022), the approximator $\mathcal{G}_\theta$ can be decomposed into the three components (Figure 2) as

$$\mathcal{G}_\theta := \mathcal{R} \circ \mathcal{A} \circ \mathcal{E}. \tag{1}$$

First, the encoder $\mathcal{E}$ takes an input function $u$ from $\mathcal{U}$ to generate the finite-dimensional encoded data in $\mathbb{R}^m$. Then, the approximator $\mathcal{A}$ is an operator approximator from the encoded data in finite dimension space $\mathbb{R}^m$ to the other finite-dimensional space $\mathbb{R}^p$. Finally, the reconstructor $\mathcal{R}$ reconstructs the output function $s(y) = \mathcal{G}(u)(y)$ with $y \in \mathcal{Y}$ using the approximated data in $\mathbb{R}^p$.

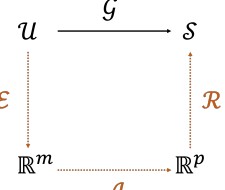

Figure 2: Diagram for the three components for operator learning.

## 3.2 DeepONet and its limitation

DeepONet can be analyzed using the above three decompositions. Assume that all the input functions $u$ are evaluated at fixed locations $\{x_j\}_{j=1}^m \subset \mathcal{X}$; they are called "sensor points." DeepONet uses an encoder as the pointwise projection $\mathcal{E}(u) = (u(x_1), u(x_2), ..., u(x_m))$ of the continuous function $u$, the so-called "sensor values" of the input function $u$. An intuitive idea is to employ a neural network that simply concatenates these $m$ sensor values and a query point $y$ as an input to approximate the target function $\mathcal{G}(u)(y)$. DeepONet, in contrast, handles $m$ sensor values and a query point $y$ separately into two subnetworks based on the universal approximation theorem for the operator (Lu et al., 2021). See Appendix B for more details. Lu et al. (2021) use the fully connected neural network for the approximator $\mathcal{A} : \mathbb{R}^m \to \mathbb{R}^p$. They referred to the composition of these two maps as branch net

$$\beta : \mathcal{U} \to \mathbb{R}^p, \beta(u) := \mathcal{A} \circ \mathcal{E}(u) \tag{2}$$

for any $u \in \mathcal{U}$. The role of branch net can be interpreted as learning the coefficient of the target function $\mathcal{G}(u)(y)$. They use one additional neural network, called trunk net $\tau$ as shown below.

$$\tau : \mathcal{Y} \to \mathbb{R}^{p+1}, \tau(y) := \{\tau_k(y)\}_{k=0}^p \tag{3}$$

for any $y \in \mathcal{Y}$. The role of trunk net can be interpreted as learning an affine space $V$ that can efficiently approximate output function space $C(\mathcal{Y}; \mathbb{R}^{d_s})$. The functions $\tau_1(y), ..., \tau_p(y)$ become the $p$-basis of vector space associated with $V$ and $\tau_0(y)$ becomes a point of $V$. By using the trunk net $\tau$, the $\tau$-induced reconstructor $\mathcal{R}$ is defined as

$$\mathcal{R}_\tau : \mathbb{R}^p \to C(\mathcal{Y}; \mathbb{R}^{d_s}), \mathcal{R}_\tau(\beta)(y) := \tau_0(y) + \sum_{k=1}^p \beta_k \tau_k(y) \tag{4}$$

where $\beta = (\beta_1, \beta_2, ..., \beta_p) \in \mathbb{R}^p$. In DeepONet, $\tau_0(y)$ is restricted to be a constant $\tau_0 \in \mathbb{R}$ that is contained in a reconstructor $\mathcal{R}$. The architecture of DeepONet is described in Figure 4 (b).

Here, the $\tau$-induced reconstructor $\mathcal{R}_\tau$ is the linear approximation of the output function space. Because the linear approximation $\mathcal{R}$ cannot consider the elements in its orthogonal complement, a priori limitation on the best error of DeepONet is explained in Lanthaler et al. (2022) as

$$\left( \int_\mathcal{U} \int_\mathcal{Y} |\mathcal{G}(u)(y) - \mathcal{R}_\tau \circ \mathcal{A} \circ \mathcal{E}(u)(y)|^2 dy d\mu(u) \right)^{\frac{1}{2}} \geq \sqrt{\sum_{k>p} \lambda_k}, \tag{5}$$

where $\lambda_1 \geq \lambda_2 \geq ...$ are the eigenvalues of the covariance operator $\Gamma_{\mathcal{G}_{\#\mu}}$ of the push-forward measure $\mathcal{G}_{\#\mu}$. Several studies point out that the slow decay rate of the lower bound leads to inaccurate approximation operator learning using DeepONet (Kovachki et al., 2021b; Hadorn, 2022; Lanthaler et al., 2022). For example, the solution operator of the advection PDEs (Seidman et al., 2022; Venturi & Casey, 2023) and of the Burgers' equation (Hadorn, 2022) are difficult to approximate when we are using the DeepONet with the small number of basis $p$.

## 3.3 Variant models of DeepONet

Several variants of DeepONet have been developed to overcome its limitation. All these models can be viewed from the perspective of parametrizing the target function as a neural network. When we think of the target network that receives $y$ as an input and generates an output $\mathcal{G}_\theta(u)(y)$, the DeepONet and its variant model can be distinguished by how information from the input function $u$ is injected into this target network $\mathcal{G}_\theta(u)$, as described in Figure 3. From this perspective, the trunk net in the DeepONet can be considered as the target network except for the final output, as shown in Figure 4 (a). The

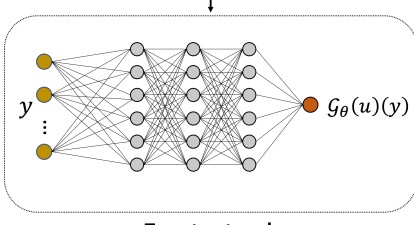

How to put information of $u \in \mathcal{U}$

**Target network**

Figure 3: The perspective target network parametrization for operator learning.

output of the branch net gives the weight between the last hidden layer and the final output.

Hadorn (2022) proposed Shift-DeepONet. The main idea is that a scale net and a shift net are used to shift and scale the input query position $y$. Therefore, it can be considered that the information of

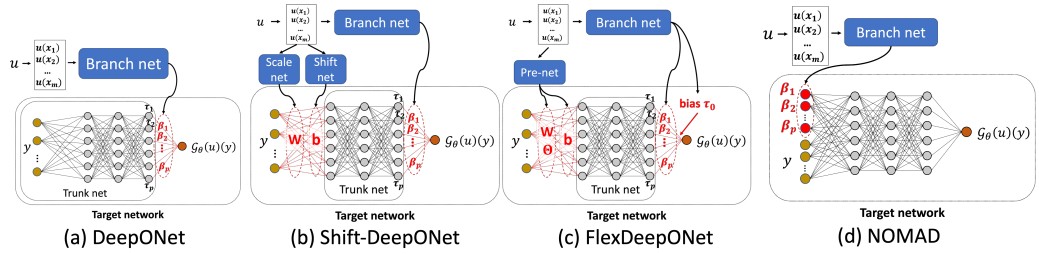

Figure 4: DeepONet and its variant models for operator learning.

input function $u$ generates the weights and bias between the input layer and the first hidden layer, as explained in Figure 4 (b).

Venturi & Casey (2023) proposed FlexDeepONet, explained in Figure 4 (c). They used the additional network, pre-net, to give the bias between the input layer and the first hidden layer. Additionally, the output of the branch net also admits the additional output $\tau_0$ to provide more information on input function $u$ at the last inner product layer.

NOMAD is recently developed by Seidman et al. (2022) to overcome the limitation of DeepONet. They devise a nonlinear output manifold using a neural network that takes the output of branch net $\{\beta_i\}_{i=1}^p$ and the query location $y$. As explained in Figure 4 (d), the target network receives information about the function $u$ as an additional input, similar to other conventional neural embedding methods (Park et al., 2019; Chen & Zhang, 2019; Mescheder et al., 2019).

These methods provide information on the input function $u$ to only a part of the target network. It is a natural idea to use a hypernetwork to share the information of input function $u$ to all parameters of the target network. We propose a general model HyperDeepONet (Figure 5), which contains the vanilla DeepONet, FlexDeepONet, and Shift-DeepONet, as a special case of the HyperDeepONet.

## 4 PROPOSED MODEL: HYPERDEEPONET

### 4.1 ARCHITECTURE OF HYPERDEEPONET

The HyperDeepONet structure is described in Figure 5. The encoder $\mathcal{E}$ and the approximator $\mathcal{A}$ are used, similar to the vanilla DeepONet. The proposed structure replaces the branch net with the hypernetwork. The hypernetwork generate all parameters of the target network. More precisely, we define the hypernetwork $h$ as

$$h_\theta : \mathcal{U} \to \mathbb{R}^p, \ h_\theta(u) := \mathcal{A} \circ \mathcal{E}(u) \qquad (6)$$

for any $u \in \mathcal{U}$. Then, $h(u) = \Theta \in \mathbb{R}^p$ is a network parameter of the target network, which is used in reconstructor for the HyperDeepONet. We define the reconstructor $\mathcal{R}$ as

$$\mathcal{R} : \mathbb{R}^p \to C(\mathcal{Y}; \mathbb{R}^{d_s}), \ \mathcal{R}(\Theta)(y) := \mathrm{NN}(y; \Theta) \qquad (7)$$

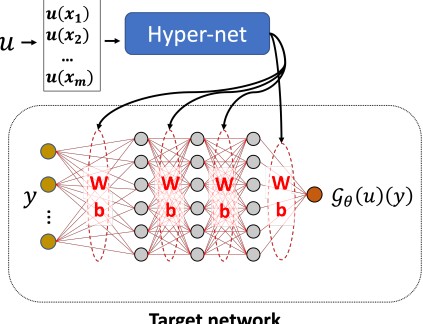

Figure 5: The proposed HyperDeepONet structure

where $\Theta = [W, b] \in \mathbb{R}^p$, and NN denotes the target network. Two fully connected neural networks are employed for the hypernetwork and target network.

Therefore, the main idea is to use the hypernetwork, which takes an input function $u$ and produces the weights of the target network. It can be thought of as a weight generator for the target network. The hypernetwork determines the all parameters of the target network containing the weights between the final hidden layer and the output layer. It implies that the structure of HyperDeepONet contains the entire structure of DeepONet. As shown in Figure 4 (b) and (c), Shift-DeepONet and FlexDeepONet can also be viewed as special cases of the HyperDeepONet, where the output of the hypernetwork determines the weights or biases of some layers of the target network. The outputs of

the hypernetwork determine the biases for the first hidden layer in the target network for NOMAD in Figure 4 (d).

## 4.2 COMPARISON ON COMPLEXITY OF DEEPONET AND HYPERDEEPONET

In this section, we would like to clarify the complexity of the DeepONet required for the approximation $\mathcal{A}$ and reconstruction $\mathcal{R}$ based on the theory in Galanti & Wolf (2020). Furthermore, we will show that the HyperDeepONet entails a relatively lower complexity than the DeepONet using the results on the upper bound for the complexity of hypernetwork (Galanti & Wolf, 2020).

### 4.2.1 NOTATIONS AND DEFINITIONS

Suppose that the pointwise projection values (sensor values) of the input function $u$ is given as $\mathcal{E}(u) = (u(x_1), u(x_2), ..., u(x_m))$. For simplicity, we consider the case $\mathcal{Y} = [-1, 1]^{d_y}$ and $\mathcal{E}(u) \in [-1, 1]^m$. For the composition $\mathcal{R} \circ \mathcal{A} : \mathbb{R}^m \to C(\mathcal{Y}; \mathbb{R})$, we focus on approximating the mapping $\mathcal{O} : \mathbb{R}^{m+d_y} \to \mathbb{R}$, which is defined as follows:

$$\mathcal{O}(\mathcal{E}(u), y) := (\mathcal{R} \circ \mathcal{A}(\mathcal{E}(u)))(y), \quad \text{for } y \in [-1, 1]^{d_y}, \mathcal{E}(u) \in [-1, 1]^m. \tag{8}$$

The supremum norm $\|h\|_\infty$ is defined as $\max_{y \in \mathcal{Y}} \|h(y)\|$. Now, we introduce the Sobolev space $\mathcal{W}_{r,n}$, which is a subset of $C^r([-1, 1]^n; \mathbb{R})$. For $r, n \in \mathbb{N}$,

$$\mathcal{W}_{r,n} := \left\{ h : [-1, 1]^n \to \mathbb{R} \quad \Big| \|h\|_r^s := \|h\|_\infty + \sum_{1 \le |\mathbf{k}| \le r} \|D^{\mathbf{k}} h\|_\infty \le 1 \right\},$$

where $D^{\mathbf{k}} h$ denotes the partial derivative of $h$ with respect to multi-index $\mathbf{k} \in \{\mathbb{N} \cup \{0\}\}^{d_y}$. We assume that the mapping $\mathcal{O}$ lies in the Sobolev space $\mathcal{W}_{r,m+d_y}$.

For the nonlinear activation $\sigma$, the class of neural network $\mathcal{F}$ represents the fully connected neural network with depth $k$ and corresponding width $(h_1 = n, h_2, \cdots, h_{k+1})$, where $W^i \in \mathbb{R}^{h_i} \times \mathbb{R}^{h_{i+1}}$ and $b_i \in \mathbb{R}^{h_{i+1}}$ denote the weights and bias of the $i$-th layer respectively.

$$\mathcal{F} := \left\{ f : [-1, 1]^n \to \mathbb{R} | f(y; [\mathbf{W}, \mathbf{b}]) = W^k \cdot \sigma(W^{k-1} \cdots \sigma(W^1 \cdot y + b^1) + b^{k-1}) + b^k \right\}$$

Some activation functions facilitate an approximation for the Sobolev space and curtail the complexity. We will refer to these functions as universal activation functions. The formal definition can be found below, where the distance between the class of neural network $\mathcal{F}$ and the Sobolev space $\mathcal{W}_{r,n}$ is defined by $d(\mathcal{F}; \mathcal{W}_{r,n}) := \sup_{\psi \in \mathcal{W}_{r,n}} \inf_{f \in \mathcal{F}} \|f - \psi\|_\infty$. Most well-known activation functions are universal activations that are infinitely differentiable and non-polynomial in any interval (Mhaskar, 1996). Furthermore, Hanin & Sellke (2017) state that the ReLU activation is also universal.

**Definition 1.** *(Galanti & Wolf, 2020) (Universal activation). The activation function $\sigma$ is called universal if there exists a class of neural network $\mathcal{F}$ with activation function $\sigma$ such that the number of parameters of $\mathcal{F}$ is $O(\epsilon^{-n/r})$ with $d(\mathcal{F}; \mathcal{W}_{r,n}) \le \epsilon$ for all $r, n \in \mathbb{N}$.*

We now introduce the theorem, which offers a guideline on the neural network architecture for operator learning. It suggests that if the entire architecture can be replaced with a fully connected neural network, large complexity should be required for approximating the target function. It also verifies that the lower bound for a universal activation function is a sharp bound on the number of parameters. First, we give an assumption to obtain the theorem.

**Assumption 1.** *Suppose that $\mathcal{F}$ and $\mathcal{W}_{r,n}$ represent the class of neural network and the target function space to approximate, respectively. Let $\mathcal{F}'$ be a neural network class representing a structure in which one neuron is added rather than $\mathcal{F}$. Then, the followings holds for all $\psi \in \mathcal{W}_{r,n}$ not contained in $\mathcal{F}$.*

$$\inf_{f \in \mathcal{F}} \|f - \psi\|_\infty > \inf_{f \in \mathcal{F}'} \|f - \psi\|_\infty.$$

For $r = 0$, Galanti & Wolf (2020) remark that the assumption is valid for 2-layered neural networks with respect to the $L^2$ norm when an activation function $\sigma$ is either a hyperbolic tangent or sigmoid function. With Assumption 1, the following theorem holds, which is a fundamental approach to identifying the complexity of DeepONet and its variants. Note that a real-valued function $g \in L^1(\mathbb{R})$ is called a bounded variation if its total variation $\sup_{\phi \in C_c^1(\mathbb{R}), \|\phi\|_\infty \le 1} \int_\mathbb{R} g(x) \phi'(x) dx$ is finite.

**Theorem 1.** *(Galanti & Wolf, 2020). Suppose that $\mathcal{F}$ is a class of neural networks with a piecewise $C^1(\mathbb{R})$ activation function $\sigma : \mathbb{R} \to \mathbb{R}$ of which derivative $\sigma'$ is bounded variation. If any non-constant $\psi \in \mathcal{W}_{r,n}$ does not belong to $\mathcal{F}$, then $d(\mathcal{F}; W_{r,n}) \leq \epsilon$ implies the number of parameters in $\mathcal{F}$ should be $\Omega(\epsilon^{-n/r})$.*

### 4.2.2 LOWER BOUND FOR THE COMPLEXITY OF THE DEEPONET

Now, we provide the minimum number of parameters in DeepONet. The following theorem presents a criterion on the DeepONet's complexity to get the desired error. It states that the number of required parameters increases when the target functions are irregular, corresponding to a small $r$. $\mathcal{F}_{\text{DeepONet}}(\mathcal{B}, \mathcal{T})$ denotes the class of function in DeepONet, induced by the class of branch net $\mathcal{B}$ and the class of trunk net $\mathcal{T}$.

**Theorem 2.** *(Complexity of DeepONet) Let $\sigma : \mathbb{R} \to \mathbb{R}$ be a universal activation function in $C^r(\mathbb{R})$ such that $\sigma$ and $\sigma'$ are bounded. Suppose that the class of branch net $\mathcal{B}$ has a bounded Sobolev norm (i.e., $\|\beta\|_r^s \leq l_1, \forall \beta \in \mathcal{B}$). Suppose any non-constant $\psi \in \mathcal{W}_{r,n}$ does not belong to any class of neural network. In that case, the number of parameters in the class of trunk net $\mathcal{T}$ is $\Omega(\epsilon^{-d_y/r})$ when $d(\mathcal{F}_{\text{DeepONet}}(\mathcal{B}, \mathcal{T}); \mathcal{W}_{r,d_y+m}) \leq \epsilon$.*

The core of this proof is showing that the inner product between the branch net and the trunk net could be replaced with a neural network that has a low complexity (Lemma 1). Therefore, the entire structure of DeepONet could be replaced with a neural network that receives $[\mathcal{E}(u), y] \in \mathbb{R}^{d_y+m}$ as input. It gives the lower bound for the number of parameters in DeepONet based on Theorem 1. The proof can be found in Appendix C.1.

The analogous results holds for variant models of DeepONet. Models such as Shift-DeepONet and flexDeepONet could achieve the desired accuracy with a small number of basis. Still, there was a trade-off in which the first hidden layer of the target network required numerous units. There was no restriction on the dimension of the last hidden layer in the target network for NOMAD, which uses a fully nonlinear reconstruction. However, the first hidden layer of the target network also had to be wide enough, increasing the number of parameters. Details can be found in Appendix C.2.

For the proposed HyperDeepONet, the sensor values $\mathcal{E}(u)$ determine the weight and bias of all other layers as well as the weight of the last layer of the target network. Due to the nonlinear activation functions between linear matrix multiplication, it is difficult to replace HyperDeepONet with a single neural network that receives $[\mathcal{E}(u), y] \in \mathbb{R}^{d_y+m}$ as input. Galanti & Wolf (2020) state that there exists a hypernetwork structure (HyperDeepONet) such that the number of parameters in the target network is $O(\epsilon^{-d_y/r})$. It implies that the HyperDeepONet reduces the complexity compared to all the variants of DeepONet.

## 5 EXPERIMENTS

In this section, we verify the effectiveness of the proposed model HyperDeepONet to learn the operators with a complicated target function space. To be more specific, we focus on operator learning problems in which the space of output function space is complicated. Each input function $u_i$ generates multiple triplet data points $(u_i, y, \mathcal{G}(u)(y))$ for different values of $y$. Except for the shallow water problem, which uses 100 training function pairs and 20 test pairs, we use 1,000 training input-output function pairs and 200 test pairs for all experiments.

For the toy example, we first consider the **identity operator** $\mathcal{G} : u_i \mapsto u_i$. The Chebyshev polynomial is used as the input (=output) for the identity operator problem. The Chebyshev polynomials of the first kind $T_l$ of degree 20 can be written as $u_i \in \{\sum_{l=0}^{19} c_l T_l(x) | c_l \in [-1/4, 1/4]\}$ with random sampling $c_l$ from uniform distribution $U[-1/4, 1/4]$. The **differentiation operator** $\mathcal{G} : u_i \mapsto \frac{d}{dx} u_i$ is considered for the second problem. Previous works handled the anti-derivative operator, which makes the output function smoother by averaging (Lu et al., 2019; 2022). Here, we choose the differentiation operator instead of the anti-derivative operator to focus on operator learning when the operator's output function space is complicated. We first sample the output function $\mathcal{G}(u)$ from the above Chebyshev polynomial of degree 20. The input function is generated using the numerical method that integrates the output function.

| Model | DeepONet | Shift | Flex | NOMAD | Hyper(ours) |
|---|---|---|---|---|---|
| Identity | 0.578±0.003 | 0.777±0.018 | 0.678±0.062 | 0.578±0.020 | **0.036±0.005** |
| Differentiation | 0.559±0.001 | 0.624±0.015 | 0.562±0.016 | 0.558±0.003 | **0.127±0.043** |

Table 1: The mean relative $L^2$ test error with standard deviation for the identity operator and the differentiation operator. The DeepONet, its variants, and the HyperDeepONet use the target network $d_y$-20-20-10-1 with *tanh* activation function. Five training trials are performed independently.

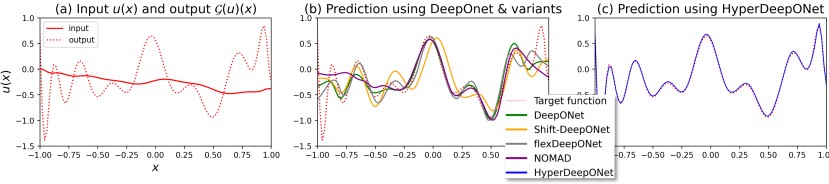

Figure 6: One test data example of differentiation operator problem.

Finally, the solution operators of PDEs are considered. We deal with two problems with the complex target function in previous works (Lu et al., 2022; Hadorn, 2022). The **solution operator of the advection equation** is considered a mapping from the rectangle shape initial input function to the solution $w(t, x)$ at $t = 0.5$, i.e., $\mathcal{G} : w(0, x) \mapsto w(0.5, x)$. We also consider the **solution operator of Burgers' equation** which maps the random initial condition to the solution $w(t, x)$ at $t = 1$, i.e., $\mathcal{G} : w(0, x) \mapsto w(1, x)$. The solution of the Burgers' equation has a discontinuity in a short time, although the initial input function is smooth. For a challenging benchmark, we consider the **solution operator of the shallow water equation** that aims to predict the fluid height $h(t, x_1, x_2)$ from the initial condition $h(0, x_1, x_2)$, i.e., $\mathcal{G} : h(0, x_1, x_2) \mapsto h(t, x_1, x_2)$ (Figure 1). In this case, the input of the target network is three dimension $(t, x_1, x_2)$, which makes the solution operator complex. Detail explanation is provided in Appendix E.

**Expressivity of target network.** We have compared the expressivity of the small target network using different models. We focus on the identity and differentiation operators in this experiment. All models employ the small target network $d_y$-20-20-10-1 with the hyperbolic tangent activation function. The branch net and the additional networks (scale net, shift net, pre-net, and hypernetwork) also use the same network size as the target network for all five models.

Table 1 shows that the DeepONet and its variant models have high errors in learning complex operators when the small target network is used. In contrast, the HyperDeepONet has lower errors than the other models. This is consistent with the theorem in the previous section that HyperDeepONet can achieve improved approximations than the DeepONet when the complexity of the target network is the same. Figure 6 shows a prediction on the differentiation operator, which has a highly complex target function. The same trends are observed when the activation function or the number of sensor points changes (Table 5) and the number of layers in the branch net and the hypernetwork vary (Figure 11).

**Same number of learnable parameters.** The previous experiments compare the models using the same target network structure. In this section, the comparison between the DeepONet and the HyperDeepONet is considered when using the same number of learnable parameters. We focus on the solution operators of the PDEs.

| | | Branch net (Hypernetwork) | Target | #Param | Rel error |
|---|---|---|---|---|---|
| Advection | DeepONet | $m$-256-256 | $d_y$-256-256-256-256-1 | 274K | 0.0046±0.0017 |
| | Hyper(ours) | $m$-70-70-70-70-70-$N_\theta$ | $d_y$-33-33-33-33-1 | 268K | 0.0048±0.0009 |
| | c-Hyper(ours) | $m$-128-128-128-128-128-1024 | $d_y$-256-256-256-256-1 | 208K | **0.0043±0.0004** |
| Burgers | DeepONet | $m$-128-128-128-128 | $d_y$-128-128-128-1 | 115K | 0.0391±0.0040 |
| | Hyper(ours) | $m$-66-66-66-66-66-$N_\theta$ | $d_y$-20-20-20-20-1 | 114K | 0.0196±0.0044 |
| | c-Hyper(ours) | $m$-66-66-66-66-66-512 | $d_y$-128-128-128-128-1 | 115K | **0.0066±0.0009** |
| Shallow | DeepONet | $m$-100-100-100-100 | $d_y$-100-100-100-100-1 | 107K | 0.0279 ± 0.0042 |
| | Hyper(ours) | $m$-30-30-30-30-$N_\theta$ | $d_y$-30-30-30-30-1 | 101K | **0.0148 ± 0.0002** |
| Shallow | DeepONet | $m$-20-20-10 | $d_y$-20-20-10-1 | 6.5K | 0.0391 ± 0.0066 |
| w/ small param | Hyper(ours) | $m$ -10-10-10-$N_\theta$ | $d_y$-10-10-10-1 | 5.7K | **0.0209 ± 0.0013** |

Table 2: The mean relative $L^2$ test error with standard deviation for solution operator learning problems. $N_\theta$ and #Param denote the number of parameters in the target network and the number of learnable parameters, respectively. Five training trials are performed independently.

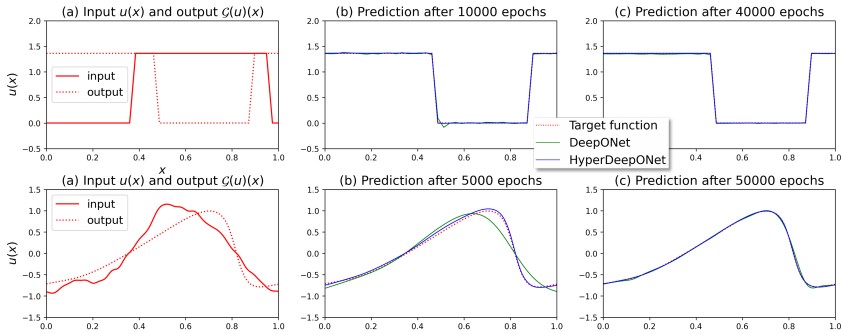

Figure 7: One test data example of prediction on the advection equation (First row) and Burgers' equation (Second row) using the DeepONet and the HyperDeepONet.

For the three solution operator learning problems, we use the same hyperparameters proposed in Lu et al. (2022) and Seidman et al. (2022) for DeepONet. First, we use the smaller target network with the larger hypernetwork for the HyperDeepONet to compare the DeepONet. Note that the vanilla DeepONet is used without the output normalization or the boundary condition enforcing techniques explained in Lu et al. (2022) to focus on the primary limitation of the DeepONet. More Details are in Appendix E. Table 2 shows that the HyperDeepONet achieves a similar or better performance than the DeepONet when the two models use the same number of learnable parameters. The HyperDeepONet has a slightly higher error for advection equation problem, but this error is close to perfect operator prediction. It shows that the complexity of target network and the number of learnable parameters can be reduced to obtain the desired accuracy using the HyperDeepONet. The fourth row of Table 2 shows that HyperDeepONet is much more effective than DeepONet in approximating the solution operator of the shallow water equation when the number of parameters is limited. Figure 7 and Figure 12 show that the HyperDeepONet learns the complex target functions in fewer epochs for the desired accuracy than the DeepONet although the HyperDeepONet requires more time to train for one epoch (Table 8).

**Scalability.** When the size of the target network for the HyperDeepONet is large, the output of the hypernetwork would be high-dimensional (Ha et al., 2017; Pawlowski et al., 2017) so that its complexity increases. In this case, the chunked HyperDeepONet (c-HyperDeepONet) can be used with a trade-off between accuracy and memory based on the chunk embedding method developed by von Oswald et al. (2020). It generates the subset of target network parameters multiple times iteratively reusing the smaller chunked hypernetwork. The c-HyperDeepONet shows a better accuracy than the DeepONet and the HyperDeepONet using an almost similar number of parameters, as shown in Table 2. However, it takes almost 2x training time and 2∼30x memory usage than the HyperDeepOnet. More details on the chunked hypernetwork are in Appendix D.

## 6 CONCLUSION AND DISCUSSION

In this work, the HyperDeepONet is developed to overcome the expressivity limitation of DeepONet. The method of incorporating an additional network and a nonlinear reconstructor could not thoroughly solve this limitation. The hypernetwork, which involves multiple weights simultaneously, had a desired complexity-reducing structure based on theory and experiments.

We only focused on when the hypernetwork and the target network is fully connected neural networks. In the future, the structure of the two networks can be replaced with a CNN or ResNet, as the structure of the branch net and trunk net of DeepONet can be changed to another network (Lu et al., 2022). Additionally, it seems interesting to research a simplified modulation network proposed by Mehta et al. (2021), which still has the same expressivity as HyperDeepONet.

Some techniques from implicit neural representation can improve the expressivity of the target network (Sitzmann et al., 2020). Using a sine function as an activation function with preprocessing will promote the expressivity of the target network. We also leave the research on the class of activation functions satisfying the assumption except for hyperbolic tangent or sigmoid functions as a future work.

ACKNOWLEDGMENTS

J. Y. Lee was supported by a KIAS Individual Grant (AP086901) via the Center for AI and Natural Sciences at Korea Institute for Advanced Study and by the Center for Advanced Computation at Korea Institute for Advanced Study. H. J. Hwang and S. W. Cho were supported by the National Research Foundation of Korea (NRF) grant funded by the Korea government (MSIT) (RS-2022-00165268) and by Institute for Information & Communications Technology Promotion (IITP) grant funded by the Korea government(MSIP) (No.2019-0-01906, Artificial Intelligence Graduate School Program (POSTECH)).

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

## A    NOTATIONS

We list the main notations in Table 3 which is not concretely described in this paper.

| | |
|---|---|
| $\mathcal{U}$ | Domain of input function |
| $\mathcal{Y}$ | Domain of output function |
| $d_u$ | Dimension of $\mathcal{U}$ |
| $d_y$ | Dimension of $\mathcal{Y}$ |
| $d_x$ | Dimension of the codomain of input function |
| $d_s$ | Dimension of the codomain of output function |
| $\{x_1, \cdots, x_m\}$ | Sensor points |
| $m$ | Number of sensor points |
| $\mathbb{R}^d$ | Euclidean space of dimension $d$ |
| $C^r(\mathbb{R})$ | Set of functions that has continuous $r$-th derivative. |
| $C(\mathcal{Y}; \mathbb{R}^{d_s})$ | Set of continuous function from $\mathcal{Y}$ to $\mathbb{R}^{d_s}$ |
| $C^r([-1, 1]^n; \mathbb{R})$ | Set of functions from $[-1, 1]^n$ to $\mathbb{R}$ whose $r$-th partial derivatives are continuous. |
| $n = O(\epsilon)$ | There exists a constant $C$ such that $n \leq C \cdot \epsilon, \forall \epsilon > 0$ |
| $n = \Omega(\epsilon)$ | There exists a constant $C$ such that $n \geq C \cdot \epsilon, \forall \epsilon > 0$ |
| $n = o(\epsilon)$ | $n/\epsilon$ converges to 0 as $\epsilon$ approaches to 0. |

Table 3: Notation

## B    ORIGINAL STRUCTURE OF DEEPONET

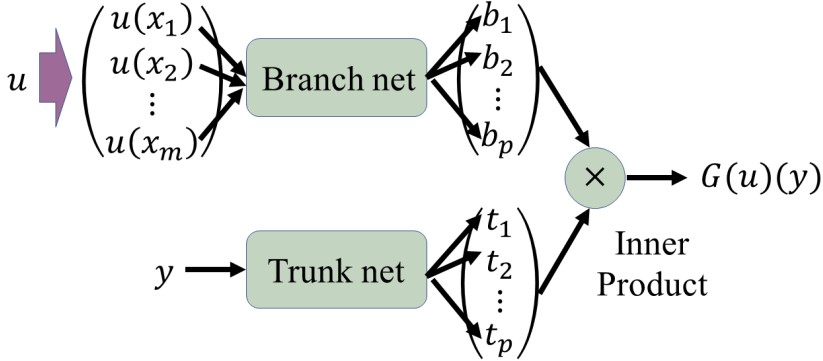

Figure 8: The structure of DeepONet

For a clear understanding of previous works, we briefly leave a description of DeepONet. In particular, we explain the structure of the unstacked DeepONet in Lu et al. (2019) which is being widely used in various experiments of the papers. Note that Figure 4(a) represents the corresponding model which is called simply DeepONet throughout this paper. The overall architecture of the model is formulated as

$$\mathcal{R}_\tau \circ \mathcal{A}_\beta(u(x_1), \cdots, u(x_m))(y) := \langle \beta(u(x_1), \cdots, u(x_m); \theta_\beta), \tau(y; \theta_\tau) \rangle,$$

where $\tau$ and $\beta$ are referred to as the trunk net and the branch net respectively. Note that $R_\tau$ and $A_\beta$ denote the reconstructor and the operator in Section 3.1. For $m$ fixed observation points $(x_1, \cdots, x_m) \in \mathcal{X}^m$, the unstacked DeepONet consists of an inner product of branch Net and trunk Net, which are fully connected neural networks. For a function $u$, the branch net receives pointwise projection values $(u(x_1), \cdots, u(x_m))$ as inputs to detect which function needs to be transformed. The trunk net queries a location $y \in \mathcal{Y}$ of interest where $\mathcal{Y}$ denotes the domain of output functions.

It was revealed that the stacked DeepONet, the simplified version of the unstacked DeepONet, is a universal approximator in the set of continuous functions. Therefore, the general structure also becomes a universal approximator which enables close approximation by using a sufficient number of parameters. Motivated by the property, we focus on how large complexity should be required for DeepONet and its variants to achieve the desired error.

## C    On complexity of DeepONet and its variants

### C.1    Proof of Theorem 2 on DeepONet complexity

The following lemma implies that the class of neural networks is sufficiently efficient to approximate the inner product.

**Lemma 1.** *For the number of basis $p \in \mathbb{N}$, consider the inner product function $\pi_p : [-1, 1]^{2p} \to \mathbb{R}$ defined by*

$$\pi_p(a_1, \cdots, a_p, b_1, \cdots, b_p) := \sum_{i=1}^{p} a_i b_i = \langle (a_1, \cdots, a_p), (b_1, \cdots, b_p) \rangle.$$

*For an arbitrary positive $t$, there exists a class of neural network $\mathcal{F}$ with universal activation $\sigma : \mathbb{R} \to \mathbb{R}$ such that the number of parameters of $\mathcal{F}$ is $O(p^{1+1/t}\epsilon^{-1/t})$ with $\inf_{f \in \mathcal{F}} \|f - \pi_p\|_\infty \leq \epsilon$.*

*Proof.* Suppose that $t$ is a positive integer and the Sobolev space $W_{2t,2}$ is well defined. First, we would like to approximate the product function $\pi_1 : [-1, 1]^2 \to \mathbb{R}$ which is defined as

$$\pi_1(a, b) = ab.$$

Note that partial derivatives $D^{\mathbf{k}}\pi_1 = 0$ for all multi-index $\mathbf{k} \in \{\mathbb{N} \bigcup \{0\}\}^2$ such that $|\mathbf{k}| \geq 2$. For a multi-index $\mathbf{k}$ with $|\mathbf{k}| = 1$, $D^{\mathbf{k}}\pi_1$ contains only one term which is either $a$ or $b$. In this case, we can simply observe that $\sum_{|\mathbf{k}|=1} \|D^{\mathbf{k}}\pi_1\|_\infty \leq 2 \cdot 1 = 2$ by the construction of the domain $[-1, 1]^2$ for $\pi_1$. And finally,

$$\|\pi_1\|_\infty \leq \|ab\|_\infty \leq \|a\|_\infty\|b\|_\infty \leq 1 \cdot 1 = 1,$$

so that a function $\pi_1/3$ should be contained in $\mathcal{W}_{r,2}$ for any $r \in \mathbb{N}$. In particular, $\pi_1/3$ lies in $\mathcal{W}_{2t,2}$ so that there exists a neural network approximation $f_{nn}$ in some class of neural network $\mathcal{F}^*$ with an universal activation function $\sigma$ such that the number of parameters of $\mathcal{F}^*$ is $O((\epsilon/3p)^{-2/2t}) = O(p^{1/t}\epsilon^{-1/t})$, and

$$\|\pi_1/3 - f_{nn}\|_\infty \leq \epsilon/3p,$$

by Definition 1. Then the neural network $3f_{nn}$ approximates the function $\pi_1$ by an error $\epsilon/p$ which can be constructed by adjusting the last weight values directly involved in the output layer of neural network $f_{nn}$.

Finally, we construct a neural network approximation for the inner product function $\pi_p$. Decompose the $2p-$dimensional inner product function $\pi_p$ into $p$ product functions $\{\text{Proj}_i(\pi_p)\}_{i=1}^{p}$ which are defined as

$$\text{Proj}_i(\pi_p) : \mathbb{R}^{2p} \to \mathbb{R}, \quad \text{Proj}_i(\pi_p)(a_1, \cdots, a_p, b_1, \cdots, b_p) := \pi_1(a_i, b_i) = a_i b_i,$$

for $\forall i \in \{1, \cdots, p\}$. Then each function $\text{Proj}_i(\pi_p)$ could be approximated within an error $\epsilon/p$ by neural network $NN_i$ which has $O(p^{1/t}\epsilon^{-1/t})$ parameters by the above discussion. Finally, by adding the last weight $[1, 1, \cdots, 1] \in \mathbb{R}^{1 \times p}$ which has input as the outputs of $p$ neural networks $\{NN_i\}_{i=1}^{p}$, we can construct the neural network approximation $NN$ of $\pi_p = \sum_{i=1}^{p} \text{Proj}_i(\pi_p)$ such that the number of parameters is $O(1 + p + p \cdot p^{1/t}\epsilon^{-1/t}) = O(p^{1+1/t}\epsilon^{-1/t})$. Class of neural network $\mathcal{F}$, which represents the structure of $NN$, satisfies the desired property.

Obviously, the statement holds for an arbitrary real $t$ which is not an integer.

$\square$

Now we assume that $\mathcal{O}$ (defined in Eq. (8)) lies in the Sobolev space $\mathcal{W}_{r,d_y+m}$. Then, we can obtain the following lemma which presents the lower bound on the number of basis $p$ in DeepONet structure. Note that we apply $L_\infty$-norm for the outputs of branch net and trunk net which are multi-dimensional vectors.

**Lemma 2.** *Let $\sigma : \mathbb{R} \to \mathbb{R}$ be a universal activation function in $C^r(\mathbb{R})$ such that $\sigma'$ is a bounded variation. Suppose that the class of branch net $\mathcal{B}$ has a bounded Sobolev norm (i.e., $\|\beta\|_r^s \leq l_1, \forall \beta \in \mathcal{B}$). Assume that supremum norm $\|\cdot\|_\infty$ of the class of trunk net $\mathcal{T}$ is bounded by $l_2$ and the number of parameters in $\mathcal{T}$ is $o(\epsilon^{-(d_y+m)/r})$. If any non-constant $\psi \in \mathcal{W}_{r,d_y+m}$ does not belong to any class of neural network, then the number of basis $p$ in $\mathcal{T}$ is $\Omega(\epsilon^{-d_y/r})$ when $d(\mathcal{F}_{DeepONet}(\mathcal{B}, \mathcal{T}); \mathcal{W}_{r,d_y+m}) \leq \epsilon$.*

*Proof.* To prove the above lemma by contradiction, we assume the opposite of the conclusion. Suppose that there is no constant $C$ that satisfies the inequality $p \geq C(\epsilon^{-d_y/r})$ for $\forall \epsilon > 0$. In other words, there exists a sequence of DeepONet which has $\{p_n\}_{n=1}^\infty$ as the number of basis with a sequence of error $\{\epsilon_n\}_{n=1}^\infty$ in $\mathbb{R}$ such that $\epsilon_n \to 0$, and satisfies

$$p_n \leq \frac{1}{n}\epsilon_n^{-d_y/r} \quad \left(\text{i.e.,} \quad p_n = o(\epsilon_n^{-d_y/r}) \quad \text{with respect to } n\right), \tag{9}$$

and

$$d(\mathcal{F}_{\text{DeepONet}}(\mathcal{B}_n, \mathcal{T}_n); \mathcal{W}_{r,d_y+m}) \leq \epsilon_n,$$

where $\mathcal{B}_n$ and $\mathcal{T}_n$ denote the corresponding class sequence of branch net and trunk net respectively. Then, we can choose the sequence of branch net $\{\beta_n : \mathbb{R}^m \to \mathbb{R}^{p_n}\}_{n=1}^\infty$ and trunk net $\left\{\tau_n : \mathbb{R}^{d_y} \to \mathbb{R}^{p_n}\right\}_{n=1}^\infty$ satisfying

$$\|\mathcal{O}(\mathcal{E}(u), y) - \pi_{p_n}(\beta_n(\mathcal{E}(u)), \tau_n(y))\|_\infty \leq 2\epsilon_n, \forall[\mathcal{E}(u), y] \in [-1, 1]^{d_y+m}, \mathcal{O} \in \mathcal{W}_{r,d_y+m}$$

for the above sequence of DeepONet by the definition of $d(\mathcal{F}_{\text{DeepONet}}(\mathcal{B}_n, \mathcal{T}_n); \mathcal{W}_{r,d_y+m})$.

Now, we would like to construct neural network approximations $f_n$ for the branch net $\beta_n$. By the assumption on the boundedness of $\mathcal{B}$, the $i$-th component $[\beta_n]_i$ of $\beta_n$ has a Sobolev norm bounded by $l_1$. In other words, $\|[\beta_n]_i/l_1\|_r^s \leq 1$ and therefore, $[\beta_n]_i/l_1$ is contained in $W_{1,m}$. Since $\sigma$ is a universal activation function, we can choose a neural network approximation $[f_n]_i$ of $[\beta_n]_i$ such that the number of parameters is $O((\epsilon_n/l_1)^{-m/r}) = O(\epsilon_n^{-m/r})$, and

$$\|[f_n]_i - [\beta_n]_i\|_\infty \leq \epsilon_n/l_1.$$

Then, $f_n = (l_1[f_n]_1, l_1[f_n]_2, \cdots, l_1[f_n]_{p_n})$ becomes neural network approximation of $\beta_n$ which has $O(p_n\epsilon_n^{-m/r})$ parameters within an error $\epsilon_n$.

Recall the target function corresponding to $m$ observation $\mathcal{E}(u) \in [-1, 1]^m$ by $\mathcal{O}(\mathcal{E}(u), \cdot) : \mathbb{R}^{d_y} \to \mathbb{R}$ which is defined in Eq. (8). Then, for $\forall \mathcal{E}(u)$, we can observe the following inequalities:

$$\|\mathcal{O}(\mathcal{E}(u), y) - \pi_{p_n}(f_n(\mathcal{E}(u)), \tau_n(y))\|_\infty$$
$$\leq \|\mathcal{O}(\mathcal{E}(u), y) - \pi_{p_n}(\beta_n(\mathcal{E}(u)), \tau_n(y))\|_\infty + \|\pi_{p_n}(\beta_n(\mathcal{E}(u)) - f_n(\mathcal{E}(u)), \tau_n(y))\|_\infty$$
$$\leq 2\epsilon_n + \epsilon_n\|\tau_n(y)\|_\infty \leq \epsilon_n(2 + l_2),$$

by the assumption on the boundedness of $\mathcal{T}$. Now we would like to consider the sequence of neural network which is an approximation of inner product between $p_n$-dimensional vector in $[-1, 1]^{p_n}$. Note the following inequality

$$\|f_n(\mathcal{E}(u))\|_\infty \leq \|f_n(\mathcal{E}(u)) - \beta_n(\mathcal{E}(u))\|_\infty + \|\beta_n(\mathcal{E}(u))\|_\infty$$
$$\leq \epsilon_n + \|\beta_n(\mathcal{E}(u))\|_r^s$$
$$\leq \epsilon_n + l_1 \leq 2l_1,$$

with $\|\tau_n\|_\infty \leq l_2$ for large $n$. It implies that $f_n(\mathcal{E}(u))/2l_1$ and $\tau_n(x)/l_2$ lie in $[-1, 1]^{p_n}$. By Lemma 1, there exists a class of neural network $\mathcal{H}_n$ such that the number of parameters is $O(p_n^{1+1/2d_yr}\epsilon_n^{-1/2d_yr})$ and,

$$\inf_{h \in \mathcal{H}_n} \|h - \pi_{p_n}\|_\infty \leq \epsilon_n$$

where $\pi_{p_n}$ is the inner product corresponding to $p_n-$dimensional vector. Choose a neural network $h_n : [-1, 1]^{2p_n} \to \mathbb{R}$ such that $\|h_n - \pi_{p_n}\|_\infty \le 2\epsilon_n$. Then, by the triangular inequality,

$$
\begin{aligned}
\|\mathcal{O}(\mathcal{E}(u), y) - 2l_1 l_2 h_n(f_n(\mathcal{E}(u))/2l_1, \tau_n(y)/l_2)\|_\infty & \\
\le \|\mathcal{O}(\mathcal{E}(u), y) - \pi_{p_n}(f_n(\mathcal{E}(u)), \tau_n(y))\|_\infty & \\
+ 2l_1 l_2 \|\pi_{p_n}(f_n(\mathcal{E}(u))/2l_1, \tau_n(y)/l_2) - h_n(f_n(\mathcal{E}(u))/2l_1, \tau_n(y)/l_2)\|_\infty & \\
\le \epsilon_n(2 + l_2) + 2l_1 l_2 (2\epsilon_n) = \epsilon_n(2 + l_2 + 4l_1 l_2). \quad (10)
\end{aligned}
$$

Finally, we compute the number of parameters which is required to implement the function $2l_1 l_2 h_n(f_n(\mathcal{E}(u))/2l_1, \tau_n(y)/l_2)$. The only part that needs further consideration is scalar multiplication. Since we need one weight to multiply a constant with one real value, three scalar multiplications

$$
\begin{aligned}
h_n(f_n(\mathcal{E}(u))/2l_1, \tau_n(y)/l_2) &\mapsto 2l_1 l_2 h_n(f_n(\mathcal{E}(u))/2l_1, \tau_n(y)/l_2), \\
f_n(\mathcal{E}(u)) &\mapsto f_n(\mathcal{E}(u))/2l_1, \quad \text{and} \quad \tau_n(x) \mapsto \tau_n(y)/l_2,
\end{aligned}
$$

require $1, p_n, p_n$-parameters respectively. Combining all the previous results with the size of trunk net, the total number of parameters is obtained in the form of

$$
O(1 + 2p_n + p_n^{1+1/2d_y r} \epsilon_n^{-1/2d_y r} + p_n \epsilon_n^{-m/r}) + o(\epsilon_n^{-(d_y+m)/r}) = o(\epsilon_n^{-(d_y+m)/r}),
$$

since the initial assumption (9) on the number of basis gives the following inequality.

$$
\begin{aligned}
p_n^{1+1/2d_y r} \epsilon_n^{-1/2d_y r} + p_n \epsilon_n^{-m/r} &\le p_n(p_n^{1/2d_y r} \epsilon_n^{-1/2d_y r} + \epsilon_n^{-m/r}) \\
&\le \frac{1}{n} \epsilon_n^{-d_y/r} (\epsilon_n^{-1/2r^2 - 1/2d_y r} + \epsilon_n^{-m/r}) \\
&\le \frac{1}{n} \epsilon_n^{-d_y/r} 2\epsilon_n^{-m/r} = \frac{2}{n} \epsilon_n^{-(d_y+m)/r}.
\end{aligned}
$$

On the one hand, the sequence of function $\{2l_1 l_2 h_n(f_n(\mathcal{E}(u))/2l_1, \tau_n(y)/l_2)\}_{n=1}^\infty$ is an sequence of approximation for $\mathcal{O}(\mathcal{E}(u), y)$ within a corresponding sequence of error $\{\epsilon_n(2 + l_2 + 4l_1 l_2)\}_{n=1}^\infty$. Denote the sequence of the class of neural networks corresponding to the sequence of the function $\{2l_1 l_2 h_n(f_n(\mathcal{E}(u))/2l_1, \tau_n(y)/l_2)\}_{n=1}^\infty$ by $\{\mathcal{F}_n\}_{n=1}^\infty$. By the assumption, Theorem 1 implies the number of parameters in $\{\mathcal{F}_n\}_{n=1}^\infty$ is $\Omega((\epsilon_n(2 + l_2 + 4l_1 l_2))^{-(d_y+m)/r}) = \Omega(\epsilon_n^{-(d_y+m)/r})$. Therefore, the initial assumption (9) would result in a contradiction so the desired property is valid. $\quad \square$

Note that the assumption on the boundedness of the trunk net could be valid if we use the bounded universal activation function $\sigma$. Using the above results, we can prove our main theorem, Theorem 2.

*Proof of Theorem 2.* Denote the number of parameters in $\mathcal{T}$ by $N_\mathcal{T}$. Suppose that there is no constant $C$ satisfies the inequality $N_\mathcal{T} \ge C\epsilon^{-d_y/r}, \forall \epsilon > 0$. That is, there exists a sequence of DeepONet with the corresponding sequence of trunk net class $\{\mathcal{T}_n\}_{n=1}^\infty$ and sequence of error $\{\epsilon_n\}_{n=1}^\infty$ such that $\epsilon_n \to 0$, and it satisfies

$$
N_{\mathcal{T}_n} < \frac{1}{n} \epsilon_n^{-d_y/r} \left( \text{i.e.,} N_{\mathcal{T}_n} = o(\epsilon_n^{-d_y/r}) \text{ with respect to } n \right).
$$

Note that the above implies $N_{\mathcal{T}_n} = o(\epsilon_n^{-(d_y+m)/r})$. On the one hand, the following inequality holds where $\mathcal{B}_n$ denotes the corresponding class sequence of branch net.

$$
d(\mathcal{F}_{\text{DeepONet}}(\mathcal{B}_n, \mathcal{T}_n); \mathcal{W}_{r, d_y+m}) \le \epsilon_n.
$$

Since $\sigma$ is bounded, $\mathcal{T}_n$ consists of bounded functions with respect to the supremum norm. Therefore, if we apply the Lemma 2 with respect to $n$, the number of basis $p_n$ should be $\Omega(\epsilon_n^{-d_y/r})$. Since $p_n$ is also the number of output dimensions for the class of trunk net $\mathcal{T}_n$, the number of parameters in $\mathcal{T}_n$ should be larger than $p_n = \Omega(\epsilon_n^{-d_y/r})$. This leads to a contradiction. $\quad \square$

Finally, we present a lower bound on the total number of parameters of DeepONet, considering the size of the branch net. Keep in mind that the proof of this theorem can be applied to other variants of DeepONet.

**Theorem 3.** *(Total Complexity of DeepONet) Let $\sigma : \mathbb{R} \to \mathbb{R}$ be a universal activation function in $C^r(\mathbb{R})$ such that $\sigma$ and $\sigma'$ are bounded. Suppose that the class of branch net $\mathcal{B}$ has a bounded Sobolev norm (i.e., $\|\beta\|_r^s \leq l_1, \forall \beta \in \mathcal{B}$). If any non-constant $\psi \in \mathcal{W}_{r,n}$ does not belong to any class of neural network, then the number of parameters in DeepONet is $\Omega(\epsilon^{-(d_y+m)/R})$ for any $R > r$ when $d(\mathcal{F}_{DeepONet}(\mathcal{B}, \mathcal{T}); \mathcal{W}_{r,d_y+m}) \leq \epsilon$.*

*Proof.* For a positive $\epsilon < l_1$, suppose that there exists a class of branch net $\mathcal{B}$ and trunk net $\mathcal{T}$ such that $d(\mathcal{F}_{\text{DeepONet}}(\mathcal{B}, \mathcal{T}); \mathcal{W}_{r,d_y+m}) \leq \epsilon$. By the boundedness of $\sigma$, there exists a constant $l_2$ which is the upper bound on the supremum norm $\| \cdot \|_\infty$ of the trunk net class $\mathcal{T}$. Let us denote the number of parameters in $\mathcal{F}_{\text{DeepONet}}$ by $N_{\mathcal{F}_{\text{DeepONet}}}$. Using the Lemma 1 to replace DeepONet's inner products with neural networks as in the inequality (10), we can construct a class of neural network $\mathcal{F}$ such that the number of parameters $\mathcal{F}$ is $O(N_{\mathcal{F}_{\text{DeepONet}}} + p^{1+1/t}\epsilon^{-1/t})$ and,

$$d(\mathcal{F}; \mathcal{W}_{r,d_y+m}) \leq (1 + l_1 l_2)\epsilon.$$

Suppose that $N_{\mathcal{F}_{\text{DeepONet}}} = o(\epsilon^{-(d_y+m)/r})$. Then, by Theorem 1, $p^{1+1/t}\epsilon^{-1/t}$ should be $\Omega(\epsilon^{-(d_y+m)/r})$. Since $t$ can be arbitrary large, the number of basis $p$ should be $\Omega(\epsilon^{-(d_y+m)/R})$ for any $R > r$. $\qquad \square$

### C.2 COMPLEXITY ANALYSIS ON VARIANTS OF DEEPONET.

We would like to verify that variant models of DeepONet require numerous units in the first hidden layer of the target network. Now we denote the class of Pre-net in Shift-DeepONet and flexDeepONet by $\mathcal{P}$. The class of Shift-DeepONet and flexDeepONet will be written as $\mathcal{F}_{\text{shift-DeepONet}}(\mathcal{P}, \mathcal{B}, \mathcal{T})$ and $\mathcal{F}_{\text{flexDeepONet}}(\mathcal{P}, \mathcal{B}, \mathcal{T})$ respectively. The structure of Shift-DeepONet can be summarized as follows. Denote the width of the first hidden layer of the target network by $w$. We define the pre-net as $\rho = [\rho_1, \rho_2] : \mathbb{R}^m \to \mathbb{R}^{w \times (d_y+1)}$ where $\rho_1 : \mathbb{R}^m \to \mathbb{R}^{w \times d_y}$ and $\rho_2 : \mathbb{R}^m \to \mathbb{R}^w$, the branch net as $\beta : \mathbb{R}^m \to \mathbb{R}^p$, and the trunk net as $\tau : \mathbb{R}^w \to \mathbb{R}^p$. The Shift-DeepONet $f_{\text{Shift-DeepONet}}(\rho, \beta, \tau)$ is defined as

$$f_{\text{Shift-DeepONet}}(\rho, \beta, \tau)(\mathcal{E}(u), y) := \pi_p(\beta(\mathcal{E}(u)), \tau(\Phi(\rho_1(\mathcal{E}(u)), y) + \rho_2(\mathcal{E}(u))))$$

where $\Phi$ is defined in Eq. (11).

We claim that it does not improve performance for the branch net to additionally output the weights on the first layer of a target network. The following lemma shows that the procedure can be replaced by a small neural network structure.

**Lemma 3.** *Consider a function $\Phi : \mathbb{R}^{d_y(w+1)} \to \mathbb{R}^w$ which is defined below.*

$$\Phi(x_1, \cdots, x_{d_y}, \cdots, x_{(d_y-1)w+1} \cdots, x_{d_y w}, y_1, \cdots, y_{d_y})$$
$$:= \left( \sum_{i=1}^{d_y} x_i y_i, \cdots, \sum_{i=1}^{d_y} x_{(d_y-1)w+i} y_i \right). \quad (11)$$

*For any arbitrary positive $t$, there exists a class of neural network $\mathcal{F}$ with universal activation $\sigma : \mathbb{R} \to \mathbb{R}$ such that the number of parameters of $\mathcal{F}$ is $O(wd_y^{1+1/t}\epsilon^{-1/t})$ with $\inf_{f \in \mathcal{F}} \|f - \Phi\|_\infty \leq \epsilon$.*

*Proof.* Using the Lemma 1, we can construct a sequence of neural network $\{f_i\}_{i=1}^w$ which is an $\epsilon$-approximation of the inner product with $O(d_y^{1+1/t}\epsilon^{-1/t})$ parameters. If we combine all of the $w$ approximations, we get the desired neural network. $\qquad \square$

Now we present the lower bound on the number of parameters for Shift-DeepONet. We derive the following theorem with an additional assumption that the class of trunk net is Lipschitz continuous. The function $\tau : \mathbb{R}^{d_y} \to \mathbb{R}^p$ is called Lipschitz continuous if there exists a constant $C$ such that

$$\|\tau(y_1) - \tau(y_2)\|_1 \leq C\|y_1 - y_2\|_1.$$

For the neural network $f$, the upper bound of the Lipschitz constant for $f$ could be obtained as $L^{k-1}\Pi_{i=1}^k \|W^i\|_1$, where $L$ is the Lipschitz constant of $\sigma$ and the norm $\| \cdot \|_1$ denotes the matrix

norm induced by vector 1-norms. We can impose constraints on the upper bound of the weights, which consequently enforce affine transformation $W^i$ to be bounded with respect to the $L^1$ norm. Therefore, we can guarantee the Lipschitz continuity of the entire neural network in this way.

We would like to remark on the validity of the weight assumptions in the theorem since the bounded assumptions of the weight may be a possible reason for increasing the number of parameters. However, the definition of Sobolev space forces all elements to have the supremum norm $\| \cdot \|_\infty$ less than 1. It may be somewhat inefficient to insist on large weights for approximating functions with a limited range.

**Theorem 4.** *Let $\sigma : \mathbb{R} \to \mathbb{R}$ be a universal activation function in $C^r(\mathbb{R})$ such that $\sigma$ and $\sigma'$ are bounded. Suppose that the class of branch net $\mathcal{B}$ and pre-net $\mathcal{P}$ has a bounded Sobolev norm (i.e., $\|\beta\|_r^s \leq l_1, \forall \beta \in \mathcal{B}$, and $\|\rho\|_r^s \leq l_3, \forall \rho \in \mathcal{P}$) and any neural network in the class of trunk net $\mathcal{T}$ is Lipschitz continuous with constant $l_2$. If any non-constant $\psi \in \mathcal{W}_{r,n}$ does not belong to any class of neural network, then the number of parameters in $\mathcal{T}$ is $\Omega(\epsilon^{-d_y/r})$ when $d(\mathcal{F}_{\text{shift-DeepONet}}(\mathcal{P}, \mathcal{B}, \mathcal{T}); \mathcal{W}_{r,d_y+m}) \leq \epsilon$.*

*Proof.* Denote the number of parameters in $\mathcal{T}$ by $N_\mathcal{T}$. Suppose that there exists a sequence of pre-net $\{\rho_n\}_{n=1}^\infty$, branch net $\{\beta_n\}_{n=1}^\infty$ and trunk net $\{\tau_n\}_{n=1}^\infty$ with the corresponding sequence of error $\{\epsilon_n\}_{n=1}^\infty$ such that $\epsilon_n \to 0$ and,

$$N_\mathcal{T} = o(\epsilon_n^{-d_y/r}), \quad \text{and} \quad \sup_{\psi \in \mathcal{W}_{r,d_y+m}} \|f_{\text{shift-DeepONet}}(\rho_n, \beta_n, \tau_n) - \psi\|_\infty \leq \epsilon_n.$$

The proof can be divided into three parts. Firstly, we come up with a neural network approximation $\rho_n^{NN}$ of $\rho_n$ of which size is $O(w \cdot \epsilon_n^{-m/r})$ within an error $\epsilon_n$. Next, construct a neural network approximation of $\Phi$ using the Lemma 3. Finally, the inner product $\pi_{p_n}(\beta_n, \tau_n)$ is replaced with a neural network as in (10) of Lemma 2.

Since all techniques such as triangular inequality are consistent with the previous discussion, we will briefly explain why additional Lipschitz continuity is required for the trunk network, and omit the details. Approximating the Pre-Net of Shift DeepOnet, which is not in DeepOnet, inevitably results in an error in the input of the trunk net. We are reluctant to allow this error to change the output of the trunk net significantly. In this situation, the Lipschitz continuity provides the desired result. $\square$

For $d_y = 1$, the additional rotation is only multiplying by 1 or $-1$. Since the weight and bias of the first layer alone can cover the scalar multiplication, flexDeepONet has the same properties as Shift-DeepONet in the above theorem.

**Theorem 5.** *Consider the case $d_y = 1$. Let $\sigma : \mathbb{R} \to \mathbb{R}$ be a universal activation function in $C^r(\mathbb{R})$ such that $\sigma$ and $\sigma'$ are bounded. Suppose that the class of branch net $\mathcal{B}$ and pre-net $\mathcal{P}$ has a bounded Sobolev norm(i.e., $\|\beta\|_r^s \leq l_1, \forall \beta \in \mathcal{B}$, and $\|\rho\|_r^s \leq l_3, \forall \rho \in \mathcal{P}$), and any neural network in the class of trunk net $\mathcal{T}$ is Lipschitz continuous with constant $l_2$. If any non-constant $\psi \in \mathcal{W}_{r,n}$ does not belong to any class of neural network, then the number of parameters in $\mathcal{T}$ is $\Omega(\epsilon^{-d_y/r})$ when $d(\mathcal{F}_{\text{flexDeepONet}}(\mathcal{B}, \mathcal{T}); \mathcal{W}_{r,d_y+m}) \leq \epsilon$.*

*Proof.* The main difference between flexDeepONet and Shift-DeepONet, which is not mentioned earlier, is that the branch net affects the bias of the output layer. However, adding the values of the two neurons can be implemented in a neural network by adding only one weight of value 1 for each neuron, so all of the previous discussion are valid. $\square$

In fact, NOMAD can be substituted with the embedding method handled by Galanti & Wolf (2020). Suppose that the branch net of NOMAD is continuously differentiable. Let's also assume that the Lipschitz constant of branch net and trunk net is bounded. We would like to briefly cite the relevant theorem here.

**Theorem 6.** *(Galanti & Wolf (2020)) Suppose that $\sigma$ is a universal activation in $C^1(\mathbb{R})$ such that $\sigma'$ is a bounded variation on $\mathbb{R}$. Additionally, suppose that there is no class of neural network that can represent any function in $\mathcal{W}_{1,d_y+m}$ other than a constant function. If the weight on the first layer of target network in NOMAD is bounded with respect to $L^1$-norm, then $d(\mathcal{N}; \mathcal{W}_{1,d_y+m}) \leq \epsilon$ implies the number of parameters in $\mathcal{N}$ is $\Omega(\epsilon^{-\min(d_y+m, 2\cdot m_y)})$ where $\mathcal{N}$ denotes the class of function contained as a target network of NOMAD.*

# D    CHUNKED EMBEDDING METHOD

The HyperDeepONet may suffer from the large complexity of the hypernetwork when the size of the target network increases. Although even a small target network can learn various operators with proper performance, a larger target network will be required for more accurate training. To take into account this case, we employ a chunk embedding method which is developed by von Oswald et al. (2020). The original hypernetwork was designed to generate all of the target network's weights so that the complexity of hypernetwork could be larger than the complexity of the target network. Such a problem can be overcome by using a hypernetwork with smaller outputs.

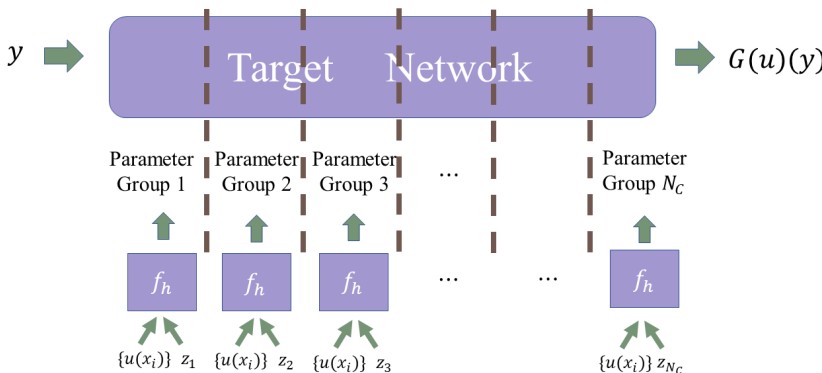

Figure 9: Chunk Embedding Method

More specifically, Figure 9 describes how the chunk embedding method reduces the number of learnable parameters. First, they partition the weights and biases of the target network. The hypernetwork then creates the weights of each parameter group by using the sensor values $\{u(x_i)\}_{i=1}^{m}$ with a latent vector $z_j$. All groups share the hypernetwork so that the complexity decreases by a factor of the number of groups. Since the latent vectors $\{z_j\}_{j=1}^{N_c}$ learn the characteristics of each group during the training period, the chunked embedding method preserves the expressivity of the hypernetwork. The chunked architecture is a universal approximator for the set of continuous functions with the existence of proper partitions (Proposition 1 in von Oswald et al. (2020)). We remark that the method can also generate the additional weights and discard the unnecessary ones when the number of the target network's parameters is not multiple of $N_C$, which is the number of group.

# E    EXPERIMENTAL DETAILS

|  | DeepONet | Shift | Flex | NOMAD | Hyper(ours) |
|---|---|---|---|---|---|
| Identity | 0.0005 | 0.0002 | 0.0005 | 0.0001 | 0.0001 |
| Differentiation | 0.0005 | 0.0002 | 0.0005 | 0.0001 | 0.0001 |
| Advection | 0.0005 | 0.0001 | 0.0001 | 0.0002 | 0.0005 |
| Burgers | 0.0001 | 0.0005 | 0.0002 | 0.0001 | 0.0001 |
| Shallow | 0.0001 | 0.0005 | - | 0.0001 | 0.0005 |

Table 4: Setting of the decay rate for each operator problem

In most experiments, we follow the hyperparameter setting in Lu et al. (2019; 2021; 2022). We use ADAM in Kingma & Ba (2015) as an optimizer with a learning rate of $1e-3$ and zero weight decay. In all experiments, an InverseTimeDecay scheduler was used, and the step size was fixed to 1. In the experiments of identity and differentiation operators, grid search was performed using the sets 0.0001, 0.0002, 0.0005, 0.001, 0.002, 0.005 for decay rates. The selected values of the decay rate for each model can be found in Table 4.

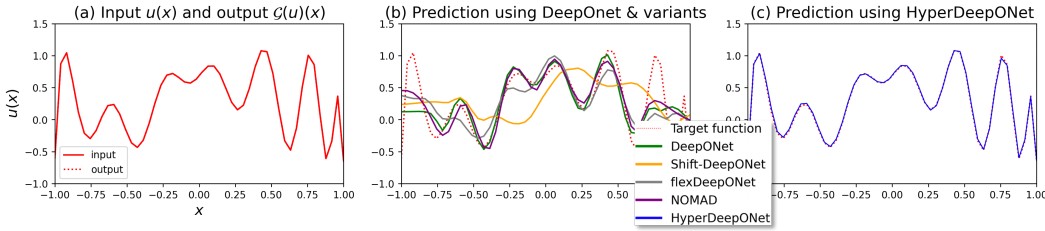

Figure 10: One test data example of differentiation operator problem.

### E.1 IDENTITY

As in the text, we developed an experiment to learn the identity operator for the 20th-order Chebyshev polynomials (Figure 10). Note that the absolute value of the coefficients of all orders is less than or equal to $1/4$. We discretize the domain $[-1, 1]$ with a spatial resolution of 50. For the experiments described in the text, we construct all of the neural networks with *tanh*. We use 1,000 training pairs and 200 pairs to validate our experiments. The batch size during the training is determined to be 5,000, which is one-tenth the size of the entire dataset.

### E.2 DIFFERENTIATION

In this experiment, we set functions whose derivatives are 20th-order Chebyshev polynomials as input functions. As mentioned above, all coefficients of the Chebyshev polynomial are between $-1/4$ and $1/4$. We use the 100 uniform grid on the domain [-1, 1]. The number of training and test samples is 1,000 and 200, respectively. We use a batch size of 10,000, which is one-tenth the size$(100, 000 = 100 \cdot 1, 000)$ of the entire dataset.

### E.3 ADVECTION EQUATION

We consider the linear advection equation on the torus $\mathbb{T} := \mathbb{R}/\mathbb{Z}$ as follows:

$$\begin{cases} \frac{\partial w}{\partial t} + c \cdot \frac{\partial w}{\partial x} = 0. \\ w(x, 0) = w_0(x), \quad x \in \mathbb{T}, \end{cases} \tag{12}$$

where $c$ is a constant which denotes the propagation speed of $w$. By constructing of the domain as $\mathbb{T}$, we implicitly assume the periodic boundary condition. In this paper, we consider the case when $c = 1$. Our goal is to learn the operator which maps $w_0(x)(= w(0, x))$ to $w(0.5, x)$. We use the same data as in Lu et al. (2022). We discretize the domain $[0, 1]$ with a spatial resolution of 40. The number of training samples and test samples is 1,000 and 200, respectively. We use the full batch for training so that the batch size is $40 \cdot 1, 000 = 40, 000$.

### E.4 BURGERS' EQUATION

We consider the 1D Burgers' equation which describes the movement of the viscous fluid

$$\begin{cases} \frac{\partial w}{\partial t} = -u \cdot \frac{\partial w}{\partial x} + \nu \frac{\partial^2 w}{\partial x^2}, & (x, t) \in (0, 1) \times (0, 1], \\ w(x, 0) = w_0(x), & x \in (0, 1), \end{cases} \tag{13}$$

where $w_0$ is the initial state and $\nu$ is the viscosity. Our goal is to learn the nonlinear **solution operator of the Burgers' equation**, which is a mapping from the initial state $w_0(x)(= w(x, 0))$ to the solution $w(x, 1)$ at $t = 1$. We use the same data of Burgers' equation provided in Li et al. (2021). The initial state $w_0(x)$ is generated from the Gaussian random field $\mathcal{N}(0, 5^4(-\Delta + 25I)^{-2})$ with the periodic boundary conditions. The split step and the fine forward Euler methods were employed to generate a solution at $t = 1$. We set viscosity $\nu$ and a spatial resolution to 0.1 and $2^7 = 128$, respectively. The size of the training sample and test sample we used are 1,000 and 200, respectively.

We take the ReLU activation function and the InverseTimeDecay scheduler to experiment with the same setting as in Lu et al. (2022). For a fair comparison, all experiments on DeepONet retained

the hyperparameter values used in Lu et al. (2022). We use the full batch so that the batch size is 1,280,000.

### E.5 SHALLOW WATER EQUATION

The shallow water equations are hyperbolic PDEs which describe the free-surface fluid flow problems. They are derived from the compressible Navier-Stokes equations. The physical conservation laws of the mass and the momentum holds in the shock of the solution. The specific form of the equation can be written as

$$
\begin{cases}
\frac{\partial h}{\partial t} + \frac{\partial}{\partial x}(hu) + \frac{\partial}{\partial y}(hv) = 0, \\
\frac{\partial(hu)}{\partial t} + \frac{\partial}{\partial x}(u^2 h + \frac{1}{2}gh^2) + \frac{\partial}{\partial y}(huv) = 0, \\
\frac{\partial(hv)}{\partial t} + \frac{\partial}{\partial y}(v^2 h + \frac{1}{2}gh^2) + \frac{\partial}{\partial x}(huv) = 0, \\
h(0, x_1, x_2) = h_0(x_1, x_2),
\end{cases}
\tag{14}
$$

for $t \in [0, 1]$ and $x_1, x_2 \in [-2.5, 2.5]$ where $h(t, x_1, x_2)$ denotes the height of water with horizontal and vertical velocity $(u, v)$. $g$ denotes the gravitational acceleration. In this paper, we aim to learn the operator $h_0(x_1, x_2) \mapsto \{h(t, x_1, x_2)\}_{t \in [1/4, 1]}$ without the information of $(u, v)$. For the sampling of initial conditions and the corresponding solutions, we directly followed the setting of Takamoto et al. (2022). The 2D radial dam break scenario is considered so that the initialization of the water height is generated as a circular bump in the center of the domain. The initial condition is generated by

$$
h(t = 0, x_1, x_2) = \begin{cases}
2.0, & \text{for } r < \sqrt{x_1^2 + x_2^2} \\
1.0, & \text{for } r \geq \sqrt{x_1^2 + x_2^2}
\end{cases}
\tag{15}
$$

with the radius $r$ randomly sampled from $U[0.3, 0.7]$. The spatial domain is determined to be a 2-dimensional rectangle $[-2.5, 2.5] \times [-2.5, 2.5]$. We use $256 = 16^2$ grids for the spatial domain. We train the models with three snapshots at $t = 0.25, 0.5, 0.75$, and predict the solution $h(t, x_1, x_2)$ for four snapshots $t = 0.25, 0.5, 0.75, 1$ on the same grid. We use 100 training samples and the batch size is determined to be 25,600.

## F  ADDITIONAL EXPERIMENTS

### F.1  COMPARISON UNDER VARIOUS CONDITIONS

The experimental results under various conditions are included in Table 5 by modifying the network structure, activation function, and number of sensor points. Although the DeepONet shows good performance in certain settings, the proposed HyperDeepONet shows good performance without dependency on the various conditions.

| Activation: ReLU, $M = 30$ | | DeepONet | Shift | Flex | NOMAD | Hyper(Ours) |
|---|---|---|---|---|---|---|
| Target network | $d_y$-30-30-30-1 | 0.16797 | 1.30852 | 1.04292 | 0.27209 | 0.02059 |
| | $d_y$-50-50-1 | 0.04822 | 1.08760 | 1.11957 | 0.21391 | 0.05562 |
| Activation: ReLU, $M = 100$ | | DeepONet | Shift | Flex | NOMAD | Hyper(Ours) |
| Target network | $d_y$-30-30-30-1 | 0.02234 | 1.08310 | 1.03741 | 0.19089 | 0.01743 |
| | $d_y$-50-50-1 | 0.07255 | 1.47373 | 1.13217 | 0.14020 | 0.04645 |
| Activation: PReLU, $M = 30$ | | DeepONet | Shift | Flex | NOMAD | Hyper(Ours) |
| Target network | $d_y$-30-30-30-1 | 0.11354 | 1.09395 | 1.03502 | 0.25651 | 0.02844 |
| | $d_y$-50-50-1 | 0.00873 | 1.14073 | 1.06947 | 0.04054 | 0.04302 |
| Activation: PReLU, $M = 100$ | | DeepONet | Shift | Flex | NOMAD | Hyper(Ours) |
| Target network | $d_y$-30-30-30-1 | 0.01035 | 1.05080 | 1.07791 | 0.16592 | 0.01083 |
| | $d_y$-50-50-1 | 0.07255 | 1.47373 | 1.13217 | 0.14020 | 0.04645 |

Table 5: The relative $L^2$ test errors for experiments on training the identity operator under various conditions

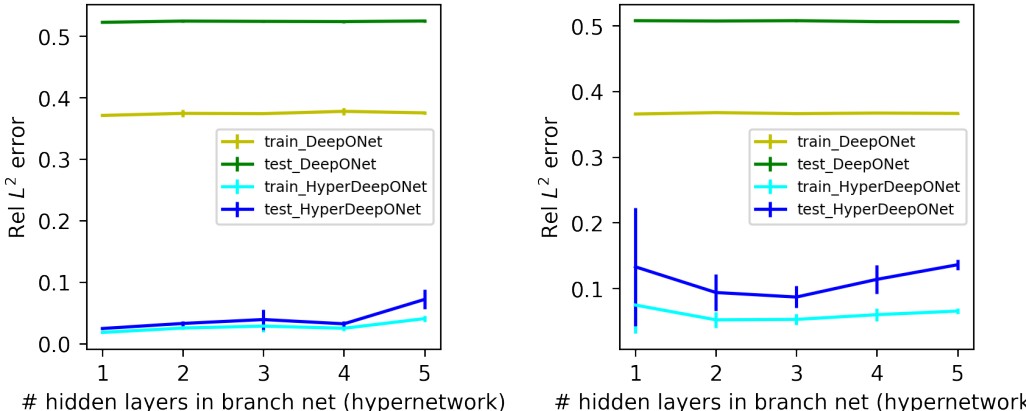

Figure 11: Varying the number of layers of branch net and hypernetwork in DeepONet and Hyper-DeepONet for identity operator problem (left) and differentiation operator problem (right).

### F.2    VARYING THE NUMBER OF LAYERS IN BRANCH NET AND HYPERNETWORK

Figure 11 compares the relative $L^2$ error of the training data and test data for the DeepONet and the HyperDeepONet by varying the number of layers in the branch net and the hypernetwork while maintaining the same small target network. Note that the bottom 150 training and test data with lower errors are selected to observe trends cleary. The training and test error for the DeepONet is not reduced despite the depth of the branch net becoming larger. This is a limitation of DeepONet's linear approximation. DeepONet approximates the operator with the dot product of the trunk net's output that approximates the basis of the target function and the branch net's output that approximates the target function's coefficient. Even if a more accurate coefficient is predicted by increasing the depth of the branch net, the error does not decrease because there is a limit to approximating the operator with a linear approximation using the already fixed trunk net.

The HyperDeepONet approximates the operator with a low test error in all cases with a different number of layers. Figure 11 shows that the training error of the HyperDeepONet remains small as the depth of the hypernetwork increases, while the test error increases. The increasing gap between the training and test errors is because of overfitting. HyperDeepONet overfits the training data because the learnable parameters of the model are more than necessary to approximate the target operator.

### F.3    COMPARISON OF HYPERDEEPONET WITH FOURIER NEURAL OPERATOR

The Fourier Neural Operator (FNO) (Li et al., 2021) is a well-known method for operator learning. Lu et al. (2022) consider 16 different tasks to explain the relative performance of the DeepONet and the FNO. They show that each method has its advantages and limitations. In particular, DeepONet has a great advantage over FNO when the input function domain is complicated, or the position of the sensor points is not uniform. Moreover, the DeepONet and the HyperDeepONet enable the inference of the solution of time-dependent PDE even in a finer time grid than a time grid used for training, e.g.the continuous-in-time solution operator of the shallow water equation in our experiment. Since the FNO is image-to-image based operator learning model, it cannot obtain a continuous solution operator over time $t$ and position $x_1, x_2$. In this paper, while retaining these advantages of DeepONets, we focused on overcoming the difficulties of DeepONets learning complex target functions because of linear approximation. Therefore, we mainly compared the vanilla DeepONet and its variants models to learn the complex target function without the result of the FNO.

Table 6 shows the simple comparison of the HyperDeepONet with the FNO for the identity operator and differentiation operator problems. Although the FNO structure has four Fourier layers, we use only one Fourier layer with 2,4,8, and 16 modes for fair comparison using similar number of parameters. The FNO shows a better performance than the HyperDeepONet for the identity operator problem. Because the FNO has a linear transform structure with a Fourier layer, the identity

| Model | HyperDeepONet (ours) | Fourier Neural Operator | | | |
|---|---|---|---|---|---|
| | | Mode 2 | Mode 4 | Mode 8 | Mode 16 |
| Identity | 0.0358 | 0.0005 | 0.0004 | 0.0003 | 0.0004 |
| Differentiation | 0.1268 | 0.8256 | 0.6084 | 0.3437 | 0.0118 |
| #Param | 15741(or 16741) | 20993 | 29185 | 45569 | 78337 |

Table 6: The relative $L^2$ test errors and the number of parameters for the identity and differentiation operator problems using HyperDeepONet and FNO with different number of modes. #Param denote the number of learnable parameters.

operator is easily approximated even with the 2 modes. In contrast, the differentiation operator is hard to approximate using the FNO with 2, 4, and 8 modes. Although the FNO with mode 16 can approximate the differentiation operator with better performance than the HyperDeepONet, it requires approximately 4.7 times as many parameters as the HyperDeepONet.

| Model | | DeepONet | Shift | Flex | NOMAD | Hyper(Ours) |
|---|---|---|---|---|---|---|
| Advection | #Param | 274K | 281K | 282K | 270K | 268K |
| | Rel error | 0.0046 | 0.0095 | 0.0391 | 0.0083 | 0.0048 |
| Burgers | #Param | 115K | 122K | 122K | 117K | 114K |
| | Rel error | 0.0391 | 0.1570 | 0.1277 | 0.0160 | 0.0196 |
| Shallow | #Param | 107K | 111K | - | 117K | 101K |
| | Rel error | 0.0279 | 0.0299 | - | 0.0167 | 0.0148 |
| Shallow | #Param | 6.5K | 8.5K | - | 6.4K | 5.6K |
| w/ small param | Rel error | 0.0391 | 0.0380 | - | 0.0216 | 0.0209 |

Table 7: The relative $L^2$ test errors and the number of parameters for the solution operators of PDEs experiments. #Param denote the number of learnable parameters. Note that the all five models use the similar number of parameters for each problem.

### F.4 PERFORMANCE OF OTHER BASELINES WITH THE SAME NUMBER OF LEARNABLE PARAMETERS

For three different PDEs with complicated target functions, we compare all the baseline methods in Table 7 to evaluate the performances. We analyze the model's computation efficiency based on the number of parameters and fix the model's complexity for each equation. All five models demonstrated their prediction abilities for the advection equation. DeepONet shows the greatest performance in this case, and other variants can no longer improve the performance. For the Burgers' equation, NOMAD and HyperDeepONet are the two outstanding algorithms from the perspective of relative test error. NOMAD seems slightly dominant to our architectures, but the two models compete within the margin of error. Furthermore, HyperDeepONet improves its accuracy using the chunk embedding method, which enlarge the target network's size while maintaining the complexity. Finally, HyperDeepONet and NOMAD outperform the other models for 2-dimensional shallow water equations. The HyperDeepONet still succeeds in accurate prediction even with a few parameters. It can be observed from Table 7 that NOMAD is slightly more sensitive to an extreme case using a low-complexity model. Because of the limitation in computing 3-dimensional rotation, FlexDeepONet cannot be applied to this problem.

Figure 13 shows the results on prediction of shallow water equations' solution operator using the DeepONet and the HyperDeepONet. The overall performance of the DeepONet is inferior to that of the HyperDeepONet, which is consistent with the result in Figure 12. In particular, the DeepONet has difficulty matching the overall circular shape of the solution when the number of parameters is small. This demonstrates the advantages of the HyperDeepONet when the computational resource is limited.

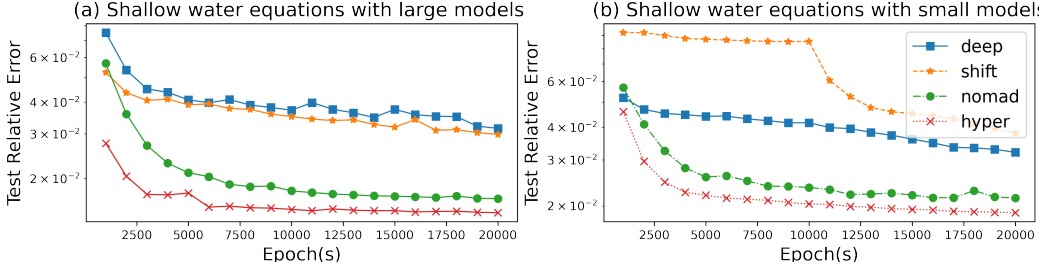

Figure 12: The test $L^2$ relative errors of four methods during training for the solution operator of shallow water equations.

|  |  | Training time (s) (per 1 epoch) | Inference time (ms) |
|---|---|---|---|
| Same target (Differentiation) | DeepONet | 1.018 | 0.883 |
|  | HyperDeepONet | 1.097 | 1.389 |
| Same #param (Advection) | DeepONet | 0.466 | 0.921 |
|  | HyperDeepONet | 0.500 | 1.912 |

Table 8: The training time and inference time for the differentiation operator problem and the solution operator of advection equation problem using DeepONet and HyperDeepONet.

### F.5 COMPARISON OF TRAINING TIME AND INFERENCE TIME

Table 8 shows the training time and the inference time for the DeepONet and the HyperDeepONet for two different operator problems. When the same small target network is employed for the DeepONet and the HyperDeepONet, the training time and inference time for the HyperDeepONet are larger than for the DeepONet. However, in this case, the time is meaningless because DeepONet does not learn the operator with the desired accuracy at all (Table 1 and Figure 6).

Even when both models use the same number of training parameters, HyperDeepONet takes slightly longer to train for one epoch than the DeepONet. However, the training complex operator using the HyperDeepONet takes fewer epochs to get the desired accuracy than DeepONet, as seen in Figure 7. This phenomenon can also be observed for the shallow water problem in Figure 12. It shows that the HyperDeepONet converges to the desired accuracy faster than any other variants of DeepONet.

The HyperDeepONet also requires a larger inference time because it can infer the target network after the hypernetwork is used to generate the target network's parameters. However, when the input function's sensor values are already fixed, the inference time to predict the output of the target function for various query points is faster than that of the DeepONet. This is because the size of the target network for HyperDeepONet is smaller than that of the DeepONet, although the total number of parameters is the same.

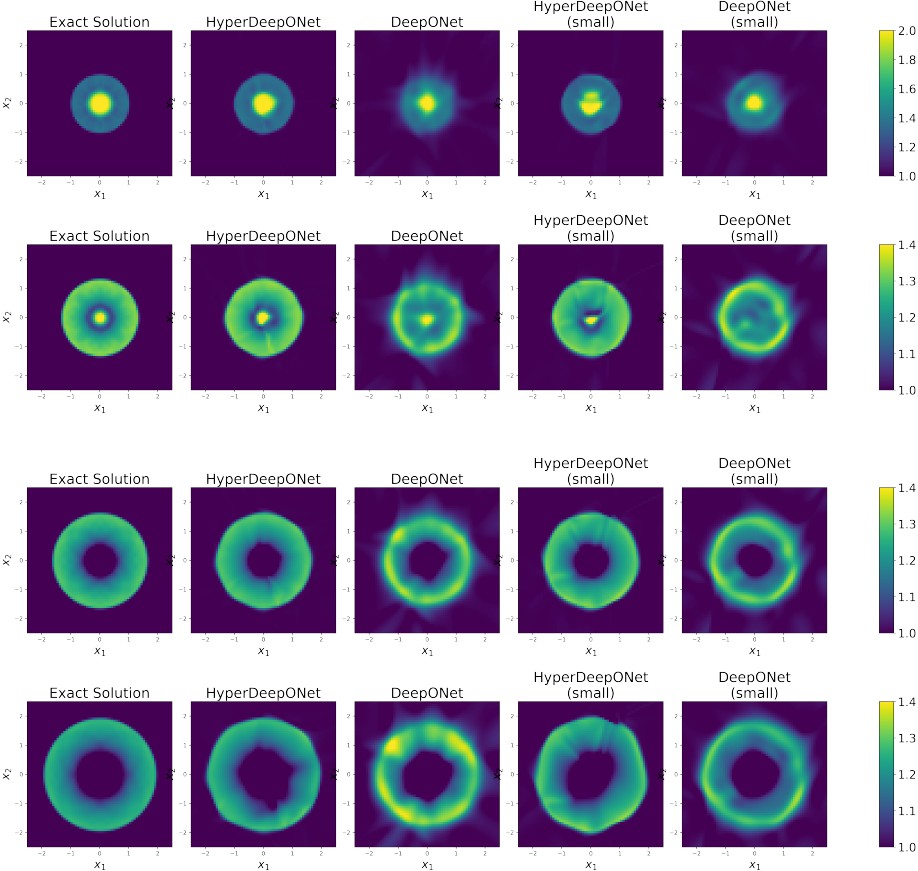

Figure 13: The examples of predictions on the solution operator of shallow water equations using the DeepONet and the HyperDeepONet. The first column represents the exact solution generated in Takamoto et al. (2022), and the other four columns denote the predicted solutions using the corresponding methods. The four rows shows the predictions $h(t, x_1, x_2)$ at four snapshots $t = [0.25, 0.5, 0.75, 1]$.

