# OpenReview forum: "HyperDeepONet: learning operator with complex target function space using the limited resources via hypernetwork"
_ICLR.cc/2023/Conference — ICLR 2023 poster_

### Official Review · Reviewer_eaJx · 2022-10-20

**Confidence:** 3
**Correctness:** 3
**Technical Novelty And Significance:** 3
**Empirical Novelty And Significance:** 2
**Recommendation:** 8

**Clarity, Quality, Novelty And Reproducibility:**

The paper requires a strong language editing because many sentences are poorly worded or unclear. I would also suggest to redo Fig. 1 to remove the upper-left component (which is very vague) and only focus on the different variants. The paper also mention related works in different points (including Section 1 and 2), I suggest to unify the discussion of related works and leave it mostly outside of the introduction.

**Strength And Weaknesses:**

I premise this review by saying I am not an expert on the field of operator learning, and I had to read the DeepONet paper to provide some judgment. In particular, I was not able to follow the theoretical derivations in Section 4 completely.

The setup reminds me of [1], where they show a Transformer network can do Bayesian inference; in both cases, the network at test time is provided with a dataset and a query point and learns to do inference by exploiting a transformer.

From what I understand, the connection made here with the hypernetwork is interesting and the paper does provide a strong theoretical / empirical motivation for replacing the original DeepONet with this variant. On the experimental part, I think more details on the actual architectures (apart from saying "fully-connected networks") should be provided.

The paper is not particularly easy to follow without reading (Lu et al., 2019) first. For example, "sensor values" are mentioned multiple times but only defined in the middle of the paper. Similarly, "p-coefficients and p-basis" is unclear in the introduction. In general, the motivation for using this architecture as compared to, e.g., simply concatenating the sensors values and the query point is not provided.

[1] https://arxiv.org/pdf/2112.10510.pdf

**Summary Of The Paper:**

The paper proposes an extension of the deep operator network (DeepONet), originally proposed in (Lu et al., 2019). The DeepONet is a way to learn operators, i.e., mappings between two functions. It takes a set of values from the first function (called sensors) and a query from the second function in input, and returns the value of the second function evaluated on the query point.

The original DeepONet framed this as learning an inner product between the representation of the sensors (branch) and the representation of the query. This paper instead proposes to parameterize directly a second neural network with the branch network, inspired by hypernetworks. They showcase on two experiments that this allows to reduce the complexity of the network. They also provide a number of theoretical insights on the reason.

**Summary Of The Review:**

My suggestion from reading this paper and the original DeepONet paper is that the contribution appears to be valuable, although the paper is not easy to follow and the experimental part appears weak if compared to the DeepONet paper. However, for the reasons mentioned above, this is only a partial evaluation and I cannot trust there are similar papers in the state-of-the-art, or the theoretical derivation has points that I was not able to evaluate carefully.

---

> ### Author Response · Authors · 2022-11-18
> **Replies to Reviewer eaJx (Part 1)**
>
> We would like to thank you for your earnest comments. We read all the comments carefully. Your comments gave us a good opportunity to review and verify the contents of the paper. We summarize your comments and respond to your concerns below.
>
> > **Q1.** The paper is not particularly easy to follow without reading (Lu et al., 2019) first. For example, "sensor values" are mentioned multiple times but only defined in the middle of the paper. Similarly, "p-coefficients and p-basis" is unclear in the introduction. In general, the motivation for using this architecture as compared to, e.g., simply concatenating the sensor values and the query point is not provided.
>
> **A.** Thanks for your constructive comments. Following your advice on a clear description, we revised the introduction part of our paper with more detailed explanations. Since the paper includes several terminologies which might me hard to understand without prior knowledge, we added a direct construction of DeepONet to Appendix B.
>
> We agree that the paper should include a more clear justification. At the beginning of Section 3.2, we include the reason why DeepONet handles $m$ sensor values and a query point $y$ separately into two subnetworks based on the universal approximation theorem for the operator. In this paper, Theorem 4.1 offers possible drawbacks of the fundamental method, which concatenates the sensor values and the query point. For a clear explanation, we include the implication of the theorem in the paragraph just above.
>
>
> > **Q2.** The setup reminds me of [1], where they show a Transformer network can do Bayesian inference; in both cases, the network at test time is provided with a dataset and a query point and learns to do inference by exploiting a transformer.
>
> **A.** We appreciate that the reviewer presented a promising intuition on the setup of our problem. [1] employed a transformer that enables learning a posterior with flexible datasets. We may construct an adaptable operator to the varying grids using the proposed structure in [1]. This method can be comparable to Variable-Input Deep Operator Networks (VIDON)[2], which considers the case where the discretization grid of the input function (sensor points) in the DeepONet changes by employing the coordinate encoder.
>
> In this paper, we focused on how efficient the proposed architecture of a neural network is to achieve the desired error for operator learning. We used the complexity and the number of parameters in neural networks, as a measure of efficiency. We analyzed the variants of DeepONet based on this perspective. It seems promising to investigate the complexity of the proposed structure in [1].

---

> > ### Comment · Reviewer_eaJx · 2022-11-30
> > **Answer to authors**
> >
> > I thank the authors for the substantive work of the review, and I apologize for the silence since re-reading everything took some time. I think the paper has substantially improved; there are still several typos and grammatical errors, but they do not prejudice the paper anymore and the ideas are more understandable. The experiments are also stronger. I have increase my evaluation accordingly.

---

> ### Author Response · Authors · 2022-11-18
> **Replies to Reviewer eaJx (Part 2)**
>
> > **Q3.** My suggestion from reading this paper and the original DeepONet paper is that the contribution appears to be valuable, although the experimental part appears weak if compared to the DeepONet paper.
>
> **A.** Thanks for pointing out important parts to help improving the paper. As you pointed out, other reviewers also commented that our paper was weak in the evaluation of our experiments. We added a more realistic example of the solution operator to show the efficacy of the proposed method, which is the shallow water equation.
>
> The shallow water equations are hyperbolic PDEs which describe the free-surface fluid flow problems. They are derived from the compressible Navier-Stokes equations. The specific form of the equation can be written as
> $$
>   \begin{cases}
>     \frac{\partial h}{\partial t} + \frac{\partial }{\partial x}(hu)+\frac{\partial }{\partial y}(hv)=0,
>     \\\ \frac{\partial (hu)}{\partial t}+\frac{\partial }{\partial x}(u^2 h + \frac{1}{2}gh^2)+\frac{\partial }{\partial y} (huv)=0,
>     \\\ \frac{\partial (hv)}{\partial t}+\frac{\partial }{\partial y}(v^2 h + \frac{1}{2}gh^2)+\frac{\partial }{\partial x} (huv)=0,
>     \\\ h(0, x_1, x_2) = h_0(x_1, x_2),
>   \end{cases}
> $$
> for $t\in[0,1]$ and $x_1, x_2 \in [-2.5, 2.5]$ where $h(t,x_1,x_2)$ denotes the height of water with horizontal and vertical velocity $(u, v)$. $g$ denotes the gravitational acceleration. In this paper, we aim to learn the operator $h_{0}(x_1, x_2) \mapsto  \left\\{h(t, x_1, x_2)\right\\}_{t\in[1/4,1]}$ without the information of $(u, v)$. For the sampling of initial conditions and the corresponding solutions, we directly followed the setting of [3]. The sample of the input function and the output function for the solution operator of the shallow water equation is in Figure 1.
>
> Table 2 shows that the HyperDeepONet also achieves a better performance than the DeepONet for the shallow water equation.  The fourth row of Table 2 shows that HyperDeepONet is much more effective than DeepONet in approximating the solution operator of the shallow water equation when the number of parameters is limited. Furthermore, Figure 12 shows that the HyperDeepONet learns the complex target functions in fewer epochs for the desired accuracy than the DeepONet. Figure 13 shows the results on prediction of shallow water equations’ solution operator using the DeepONet and the HyperDeepONet. The overall performance of the DeepONet is inferior to that of the HyperDeepONet. In particular, the DeepONet has difficulty matching the overall circular shape of the solution when the number of parameters is small. This demonstrates the advantages of the HyperDeepONet when the computational resource is limited.
>
> > **Q4.** The paper requires a strong language editing because many sentences are poorly worded or unclear. The paper is not easy to follow
>
> **A.** Thanks for your comments on the writing. We read the whole paper and revised the sentences more clearly. You can see it in the updated manuscript.
>
> > **Q5.** On the experimental part, I think more details on the actual architectures (apart from saying "fully-connected networks") should be provided.
>
> **A.** As you point out, there seems to be a lack of explanation in the HyperDeepONet model. We added a detailed description of HyperDeepONet in section 4.1.
>
> > **Q6.** I would also suggest to redo Fig. 1 to remove the upper-left component (which is very vague) and only focus on the different variants.
>
> **A.** Thank you for the valuable comment. It seems that your suggestion is related to Figure 2, not Figure 1 of the previous paper. As you suggested, the figure is split into two figures to avoid confusion for the readers. In the revised paper, Figure 3 is an explanation of the target network perspective, and Figure 4 focuses on DeepONet and its variant models as you suggested.
>
> > **Q7.** The paper also mention related works in different points (including Section 1 and 2), I suggest to unify the discussion of related works and leave it mostly outside of the introduction.
>
> **A.** Thank you so much for giving us a good point. As you pointed out, several papers are cited repeatedly in Section 1 (introduction) and Section 2 (related work). It would be more natural to cite and explain some papers in Section 2 rather than Section 1. You can check the changes made reflecting your points in the revised version of the manuscript.
>
>
> *[1] Müller, Samuel, et al. "Transformers Can Do Bayesian Inference." _arXiv preprint arXiv:2112.10510_ (2021).*
>
> *[2] Prasthofer, Michael, Tim De Ryck, and Siddhartha Mishra. "Variable-Input Deep Operator Networks." arXiv preprint arXiv:2205.11404 (2022).*
>
> *[3] Takamoto, Makoto, et al. "PDEBENCH: An Extensive Benchmark for Scientific Machine Learning." _arXiv preprint arXiv:2210.07182_ (2022).*

---

> ### Author Response · Authors · 2022-11-30
> **Reminder**
>
> Dear Reviewer eaJx,
>
> we would like to friendly remind you that we submitted our response to your review comments two weeks ago. The discussion period (stage 2) will ends in two weeks. Please let us know if our response resolves your concerns and we appreciate it if you could give us any feedback. We thank you again for your valuable comments and suggestions.

---

### Official Review · Reviewer_8Dyr · 2022-10-25

**Confidence:** 3
**Correctness:** 4
**Technical Novelty And Significance:** 3
**Empirical Novelty And Significance:** 3
**Recommendation:** 8

**Clarity, Quality, Novelty And Reproducibility:**

The paper is quite clear, and the background, problem setup, and related methods are well explained. While the proposed methodological improvements are not particularly novel - they ultimately amount to small changes in the neural network parameterization - it is well motivated and comes with theoretical results. Code is provided for reproducing experiments.

The paper's significance and potential impact could be limited by the niche nature of this field. While the problem setup is clear from a formalism perspective, one suggestion that could make the paper better motivated and easier to read is to provide a simple running example throughout the paper about the problem setup and why these operator learning problems are important.

Minor: sometimes the word "complex" is used in a confusing manner where the context lends itself to be interpreted as "complex as in $\mathbb{C}$" when the authors seem to mean "complex as in complicated". For example, the second sentence of Section 5: "To be more specific, we focus on operator learning problems in which the space of output function space is complex."


**Strength And Weaknesses:**

Strengths:
- Well-explained problem setup and background
- Method is well motivated
- Method comes with theoretical results for both upper and lower bounds (Theorem 1 and 2)
- Well-rounded empirical study including metrics, visualization of predictions, and several baselines and ablations

Weaknesses:
- Evaluation is very small-scale, consisting of largely of synthetic data and simple PDEs, and is hard to tell whether the proposed method can be of practical significance on more interesting problems. While results look promising, there are ultimately very few quantitative results and it is hard to evaluate the efficacy of the method.
- A discussion of the tradeoffs of the proposed method is missing (i.e. potential weaknesses). For example, when the network is more complicated as in HyperDeepONet, how is the computational speed compared to a parameter-matched DeepONet?


**Summary Of The Paper:**

This paper builds on the deep operator network (DeepONet) for learning operators between function spaces, for examples for PDE solving. This paper proposes HyperDeepONet which replaces the network with a hypernetwork, in other words making the target function input-dependent. This method is shown theoretically to be more parameter efficient and improves on related baselines on synthetic experiments.


**Summary Of The Review:**

This paper provides an improvement to the deep operator network approach to operator learning, which is well motivated and has promising empirical results.

----------
Post rebuttal:
The authors have provided new content, figures, and experiments addressing my concerns and overall improved the presentation of the paper. I am increasing my score.

---

> ### Author Response · Authors · 2022-11-18
> **Replies to Reviewer 8Dyr (Part 1)**
>
> We appreciate your constructive and insightful comments. We read all the comments carefully. Your comments gave us a good opportunity to review and verify the contents of the paper. We summarize your comments and respond to your concerns below.
>
>
> > **Q1.** Evaluation is very small-scale, consisting of largely of synthetic data and simple PDEs, and is hard to tell whether the proposed method can be of practical significance on more interesting problems. While results look promising, there are ultimately very few quantitative results and it is hard to evaluate the efficacy of the method.
>
> **A.** Thank you for your suggestion for improving the evaluation of our experiments.  We agree on this point and added a more practical and interesting experiment, which is the solution operator of the shallow water equation.
>
> The shallow water equations are hyperbolic PDEs which describe the free-surface fluid flow problems. They are derived from the compressible Navier-Stokes equations. The specific form of the equation can be written as
> $$
>   \begin{cases}
>     \frac{\partial h}{\partial t} + \frac{\partial }{\partial x}(hu)+\frac{\partial }{\partial y}(hv)=0,
>     \\\ \frac{\partial (hu)}{\partial t}+\frac{\partial }{\partial x}(u^2 h + \frac{1}{2}gh^2)+\frac{\partial }{\partial y} (huv)=0,
>     \\\ \frac{\partial (hv)}{\partial t}+\frac{\partial }{\partial y}(v^2 h + \frac{1}{2}gh^2)+\frac{\partial }{\partial x} (huv)=0,
>     \\\ h(0, x_1, x_2) = h_0(x_1, x_2),
>   \end{cases}
> $$
> for $t\in[0,1]$ and $x_1, x_2 \in [-2.5, 2.5]$ where $h(t,x_1,x_2)$ denotes the height of water with horizontal and vertical velocity $(u, v)$. $g$ denotes the gravitational acceleration. In this paper, we aim to learn the operator $h_{0}(x_1, x_2) \mapsto  \left\\{h(t, x_1, x_2)\right\\}_{t\in[1/4,1]}$ without the information of $(u, v)$. For the sampling of initial conditions and the corresponding solutions, we directly followed the setting of [1]. The sample of the input function and the output function for the solution operator of the shallow water equation is in Figure 1.
>
> Table 2 shows that the HyperDeepONet also achieves a better performance than the DeepONet for the shallow water equation.  The fourth row of Table 2 shows that HyperDeepONet is much more effective than DeepONet in approximating the solution operator of the shallow water equation when the number of parameters is limited. Furthermore, Figure 12 shows that the HyperDeepONet learns the complex target functions in fewer epochs for the desired accuracy than the DeepONet. Figure 13 shows the results on prediction of shallow water equations’ solution operator using the DeepONet and the HyperDeepONet. The overall performance of the DeepONet is inferior to that of the HyperDeepONet. In particular, the DeepONet has difficulty matching the overall circular shape of the solution when the number of parameters is small. This demonstrates the advantages of the HyperDeepONet when the computational resource is limited.
>
> Furthermore, we added several experimental results to show the advantages of our model in the text and appendix (e.g., comparison with FNO, test error during training, etc.).

---

> ### Author Response · Authors · 2022-11-18
> **Replies to Reviewer 8Dyr (Part 2)**
>
> > **Q2.** A discussion of the tradeoffs of the proposed method is missing (i.e. potential weaknesses). For example, when the network is more complicated as in HyperDeepONet, how is the computational speed compared to a parameter-matched DeepONet?
>
> **A.** Thank you so much for your good point. We added a discussion on two aspects of the trade-off of our model. First, HyperDeepONet has a weakness in the computational speed compared to DeepONet. Table 8 shows the training time and the inference time for the DeepONet and the HyperDeepONet for two different operator problems. When the same small target network is employed for the DeepONet and the HyperDeepONet, the training time and inference time for the HyperDeepONet is larger than for the DeepONet. However, in this case, the time is meaningless because DeepONet does not learn the operator with the desired accuracy at all (Table 1 and Figure 6).
>
> Even when both models use the same number of training parameters, HyperDeepONet takes slightly longer to train for one epoch than the DeepONet. However, the training complex operator using the HyperDeepONet takes fewer epochs to get the desired accuracy than DeepONet, as seen in Figure 7. This phenomenon can also be observed for the shallow water problem in Figure 12. It shows that the HyperDeepONet converges to the desired accuracy faster than any other variants of DeepONet.
>
> The HyperDeepONet also requires a larger inference time because it can infer the target network after the hypernetwork is used to generate the target network’s parameters. However, when the input function’s sensor values are already fixed, the inference time to predict the output of the target function for various query points is faster than that of the DeepONet. This is because the size of the target network for HyperDeepONet is smaller than that of the DeepONet, although the total number of parameters is the same.
>
> Second, the output of the hypernetwork would be high-dimensional so that its complexity increases when the size of the target network for the HyperDeepONet is large. In this case, the chunked HyperDeepONet (c-HyperDeepONet) can be used with a trade-off between accuracy and memory based on the chunk embedding method developed by [2]. It generates the subset of target network parameters multiple times iteratively reusing the smaller chunked hypernetwork. The c-HyperDeepONet shows a better accuracy than the DeepONet and the HyperDeepONet using an almost similar number of parameters, as shown in Table 2. However, it takes almost 2x training time and 2∼30x memory usage than the HyperDeepOnet. More details on the chunked hypernetwork are in Appendix D.
>
>
>
> > **Q3.** The paper's significance and potential impact is limited by the niche nature of this field. While the problem setup is clear from a formalism perspective, one suggestion that could make the paper better motivated and easier to read is to provide a simple running example throughout the paper about the problem setup and why these operator learning problems are important.
>
> **A.** Thanks for your constructive suggestions. As you pointed out, the problem setup with formulas can be difficult to understand for readers who are unfamiliar with operator learning. To this end, we explained why operator learning is important with an example of the shallow water equation after the problem setup of the operator learning. Furthermore, the example of the input function and the output function of the operator is included in Figure 1 to aid the readers' understanding.
>
> > **Q4.** Minor: sometimes the word "complex" is used in a confusing manner where the context lends itself to be interpreted as "complex as in $\mathbb{C}$" when the authors seem to mean "complex as in complicated"
>
> **A.** Thank you for pointing out an important point that may confuse the readers. We read and corrected the contexts where the word 'complex' could be interpreted as '$\mathbb{C}$' rather than 'complicated'. You can see it in the updated manuscript.
>
> *[1] Takamoto, Makoto, et al. "PDEBENCH: An Extensive Benchmark for Scientific Machine Learning." _arXiv preprint arXiv:2210.07182_ (2022).*
>
> *[2] Von Oswald, Johannes, et al. "Continual learning with hypernetworks." _arXiv preprint arXiv:1906.00695_ (2019).*

---

> > ### Comment · Reviewer_8Dyr · 2022-11-23
> > **Response**
> >
> > Thank you for the detailed responses! The authors have made many improvements to the presentation of the paper, particularly including a running example on a real problem with new empirical results. The improvements have largely addressed my original concerns, and I will increase my score after some more minor comments and questions:
> >
> > - Incorporating the example in Section 3.1 and Figure 1 substantially helps the reader understand the problem setup. I was initially confused about the role of the time axis in the example. My understanding was that some related methods such at FNO work only for one output time $t$ (that is a different network must be trained for each output time). The way Figure 1 is drawn makes it look similar, that two separate operators were being learned for $t=0.5$ and $t=1$. Reading the example and the details of the problem setup however makes it more clear that $t$ is included in the axis of the operator output so that it works for all timestamps (could you confirm that this is true?)
> > - One little addition that could help make this sort of thing very clear is connecting the text in the Section 3.1 example back to the notation. In particular, specifying $d_x=2, d_y=3, d_u=1, d_s=1$ if my interpretation is correct
> > - Minor: there are a couple more places where "complex functions" are still used, e.g. (1) last before before bullets in the intro "involves a complex target function space" (2) red text below Figure 6.  (I realize the authors can't update the draft, just pointing some minor copy edits for the future)
> > - Very minor: some of the new additions use the wrong left quote

---

> > > ### Author Response · Authors · 2022-11-23
> > > **Thank you for your comments and questions**
> > >
> > > We sincerely thank you for your valuable comments and questions. As you understand correctly, our model has a time $t$ variable as one axis of the operator's output function for the shallow water equation, so the model can predict continuously for all timestamps. Furthermore, the following explanation was added to Appendix F.3  in the revised manuscript, which compares FNO and our model.
> > > > "Moreover, the DeepONet and the HyperDeepONet enable the inference of the solution of time-dependent PDE even in a finer time grid than a time grid used for training, e.g. the continuous-in-time solution operator of the shallow water equation in our experiment."
> > >
> > > Therefore, the solution operator we are trying to learn is $\mathcal{G}:h(0,x_1,x_2)\mapsto h(t, x_1, x_2)$, which is the case of $d_x=2$, $d_y=3$, $d_u=1$, $d_s=1$ according to your interpretation.
> > >
> > > As you pointed out, Figure 1 seems misleading as it trains two different operators $\mathcal{G}:h(0,x_1,x_2)\mapsto h(t=0.5, x_1, x_2)$ and $\mathcal{G}:h(0,x_1,x_2)\mapsto h(t=1, x_1, x_2)$ separately. When the final revision is possible in the future, Figure 1 will be modified so that there is no confusion. Thank you for bringing up something the authors might be confused about.
> > >
> > > We would like to apologize for the reuse of the confusing words ‘complex’ in our manuscript,
> > > Following the reviewer’s suggestions, we will revise our manuscript with more careful attention to the expression. We also appreciate pointing out the incorrect left quote in Section 3.2. We will revise our quotation marks for the future version.

---

> > > > ### Comment · Reviewer_8Dyr · 2022-11-23
> > > > **Thanks**
> > > >
> > > > Thanks for the clarifications! This is a very solid paper, especially after the substantial amount of revision and new content during the rebuttal phase. I am increasing my score.

---

### Official Review · Reviewer_LzY8 · 2022-10-31

**Confidence:** 2
**Correctness:** 3
**Technical Novelty And Significance:** 3
**Empirical Novelty And Significance:** 2
**Recommendation:** 8

**Clarity, Quality, Novelty And Reproducibility:**

Clarity: Fair. The writing of this paper could definitely be further improved.

Quality: Good. The idea is interesting, along with theoretical and empirical evidence to support it.

Novelty: I'm not an expert in this area, so I cannot fairly evaluate the novelty of this paper.

Reproducibility: Good, the authors provide the code of this paper.

**Strength And Weaknesses:**

**Pros:**

* The motivation is clear, and the idea of leveraging a hypernetwork to generate parameters of the target network is interesting.
* The authors analyze DeepONet with its variants from the perspective of: "how information from the input function $u$ is injected into the target network $\mathcal{G}_{\theta}(u)$" as illustrated in Figure 2. It is quite clear and makes connections among these methods, including HyperDeepONet.
* Thorough theoretical analysis of the complexity is provided (but I cannot follow this part).

**Cons:**

* The writing of this paper could be further polished. Many typos also exist.

* About the experiment section: I do not see the results of FNO [1], which also belongs to operator learning. Could the authors explain the reason? Moreover, A PDE benchmark [2] has been proposed recently. It would be better to provide some experimental results on it (the second half does not affect my evaluation since it was proposed recently).
* The scalability of the proposed method seems to be an issue. This paper mainly considers the complexity of the target network. It would be better to consider the complexity of the branch network (also the hypernet) as well in the proposed method (i.e., the complexity of the Hyper will increase as that of the target network increases).

**Question:**

* About the experiment section,
  * in Figure 7 and 8, could the authors explain why increasing the number of layers in the branch net does not reduce the $L^2$ error of DeepONet. For the HyperDeepONet, why it may even increase the error? What is the intuition behind this phenomenon?
  * when considering the whole complexity (i.e., the same total Params in table 2), the performance of Hyper seems to be inferior to that of DeepONet. How about the performance of other baselines?
* The authors mention that real-time prediction on resource-constrained hardware is crucial and challenging, so for each method, how about the training time, and inference time to obtain a desired prediction?

**Minor:**

* Please explicitly define the abbreviation when they first appear, e.g., PDEs on Page 1.

* "shows" -> "show"; "small" -> "a small" in point 3 of main contributions.
* "solving" -> "solve"; "nonlinear" -> "a nonlinear" in the second paragraph of related work.
* "considered" -> "considered as" in section 3.3.

[1] Fourier neural operator for parametric partial differential equations.

[2] PDEBench: An Extensive Benchmark for Scientific Machine Learning.

**Summary Of The Paper:**

This paper targets a cutting-edge research problem that tries to use deep neural networks to solve partial differential equations (PDEs) through operator learning, with the potential to achieve faster and/or more accurate predictions for complex physical dynamics than traditional numerical methods. Although previously proposed methods (e.g., DeepONet and its variants) achieve some success, due to the limitation of linear approximation investigated by recent work, they typically need a large number of parameters and computational costs to learn the complex operators. In order to solve the problem, the authors propose HyperDeepONet, which learns complex operators with a target network whose parameters are generated by a hypernetwork (conditioned on the input function $u(\cdot)$). Both theoretical and empirical evidence is provided to demonstrate the lower complexity and effectiveness of the proposed method.

**Summary Of The Review:**

Overall, this paper focuses on a significant research problem and proposes an interesting method to make learning complex operators with lower complexity (e.g., fewer parameters) possible. Well-rounded theory and empirical results are provided. I believe this work could be further improved with better writing and further evaluation. Based on the current version, I recommend a borderline acceptance.

**Update on 04 Nov after reading two other reviews:** I agree with the other two reviewers, and my main concerns about this work are: a) the writing is not good (*eaJx*); b) the evaluation is not strong (*8Dyr*). If the authors could resolve these two concerns and update the draft accordingly before 18 Nov, I would be happy to recommend this work.

---

> ### Author Response · Authors · 2022-11-18
> **Replies to Reviewer LzY8 (Part 1)**
>
> We sincerely appreciate your valuable and thoughtful comments. We read all the comments carefully. You gave us many meaningful opinions, ideas, and questions. The comments are indeed helpful to improve our paper. We summarize your comments and respond to your concerns below.
>
> > **Q1.** About the experiment section: I do not see the results of FNO [1], which also belongs to operator learning. Could the authors explain the reason?
>
> **A.** As you pointed out, Fourier neural operator (FNO) [1] is a well-known operator learning method alongside deep operator network (DeepONet). [3] consider 16 different tasks to explain the relative performance of the DeepONet and the FNO. They show that each method has its advantages and limitations. In particular, DeepONet has a great advantage over FNO when the input function domain is complicated, or the position of the sensor points is not uniform. Moreover, the DeepONet and the HyperDeepONet enable the inference of the solution of time-dependent PDE even in a finer time grid than a time grid used for training, e.g.the continuous-in-time solution operator of the shallow water equation in our experiment. Since the FNO is image-to-image based operator learning model, it cannot obtain a continuous solution operator over time $t$ and position $x_1, x_2$. In this paper, while retaining these advantages of DeepONets, we focused on overcoming the difficulties of DeepONets learning complex target functions because of linear approximation. Therefore, we mainly compared the vanilla DeepONet and its variants models to learn the complex target function without the result of the FNO.
>
> Table 6 shows the simple comparison of the HyperDeepONet with the FNO for the identity operator and differentiation operator problems. Although the FNO structure has four Fourier layers, we use only one Fourier layer with 2,4,8, and 16 modes for fair comparison using similar number of parameters. The FNO shows a better performance than the HyperDeepONet for the identity operator problem. Because the FNO has a linear transform structure with a Fourier layer, the identity operator is easily approximated even with the 2 modes. In contrast, the differentiation operator is hard to approximate using the FNO with 2, 4, and 8 modes. Although the FNO with mode 16 can approximate the differentiation operator with better performance than the HyperDeepONet, it requires approximately 4.7 times as many parameters as the HyperDeepONet.

---

> ### Author Response · Authors · 2022-11-18
> **Replies to Reviewer LzY8 (Part 2)**
>
> > **Q2.** Moreover, A PDE benchmark [2] has been proposed recently. It would be better to provide some experimental results on it
>
> **A.** Thank you for your suggestions to improve the evaluations of our experiments. We didn't know the paper [2] because it was recently published. There were various practical and interesting benchmarks for scientific machine learning. Among them, we chose the shallow water equation that has a complex target function suitable for showing the efficacy of the proposed method.
>
> The shallow water equations are hyperbolic PDEs which describe the free-surface fluid flow problems. They are derived from the compressible Navier-Stokes equations. The specific form of the equation can be written as
> $$
>   \begin{cases}
>     \frac{\partial h}{\partial t} + \frac{\partial }{\partial x}(hu)+\frac{\partial }{\partial y}(hv)=0,
>     \\\ \frac{\partial (hu)}{\partial t}+\frac{\partial }{\partial x}(u^2 h + \frac{1}{2}gh^2)+\frac{\partial }{\partial y} (huv)=0,
>     \\\ \frac{\partial (hv)}{\partial t}+\frac{\partial }{\partial y}(v^2 h + \frac{1}{2}gh^2)+\frac{\partial }{\partial x} (huv)=0,
>     \\\ h(0, x_1, x_2) = h_0(x_1, x_2),
>   \end{cases}
> $$
> for $t\in[0,1]$ and $x_1, x_2 \in [-2.5, 2.5]$ where $h(t,x_1,x_2)$ denotes the height of water with horizontal and vertical velocity $(u, v)$. $g$ denotes the gravitational acceleration. In this paper, we aim to learn the operator $h_{0}(x_1, x_2) \mapsto  \left\\{h(t, x_1, x_2)\right\\}_{t\in[1/4,1]}$ without the information of $(u, v)$. For the sampling of initial conditions and the corresponding solutions, we directly followed the setting of [2]. The sample of the input function and the output function for the solution operator of the shallow water equation is in Figure 1. More details are in Appendix E.5.
>
> Table 2 shows that the HyperDeepONet also achieves a better performance than the DeepONet for the shallow water equation.  The fourth row of Table 2 shows that HyperDeepONet is much more effective than DeepONet in approximating the solution operator of the shallow water equation when the number of parameters is limited. Furthermore, Figure 12 shows that the HyperDeepONet learns the complex target functions in fewer epochs for the desired accuracy than the DeepONet. Figure 13 shows the results on prediction of shallow water equations’ solution operator using the DeepONet and the HyperDeepONet. The overall performance of the DeepONet is inferior to that of the HyperDeepONet. In particular, the DeepONet has difficulty matching the overall circular shape of the solution when the number of parameters is small. This demonstrates the advantages of the HyperDeepONet when the computational resource is limited.
>
> > **Q3.** The scalability of the proposed method seems to be an issue. This paper mainly considers the complexity of the target network. It would be better to consider the complexity of the branch network (also the hypernet) as well in the proposed method (i.e., the complexity of the Hyper will increase as that of the target network increases).
>
> **A.** Thank you for pointing out the scalability of the proposed HyperDeepONet, which is the most important part of our model. It was also our future work that was included in the conclusion section. We added experiments and explanations to address the scalability of HyperDeepONet.
>
> When the size of the target network for the HyperDeepONet is large, the output of the hypernetwork would be high-dimensional so that its complexity increases. In this case, the chunked HyperDeepONet (c-HyperDeepONet) can be used with a trade-off between accuracy and memory based on the chunk embedding method developed by [4]. It generates the subset of target network parameters multiple times iteratively reusing the smaller chunked hypernetwork. The c-HyperDeepONet shows a better accuracy than the DeepONet and the HyperDeepONet using an almost similar number of parameters, as shown in Table 2. However, it takes almost 2x training time and 2∼30x memory usage than the HyperDeepOnet. More details on the chunked hypernetwork are in Appendix D.

---

> ### Author Response · Authors · 2022-11-18
> **Replies to Reviewer LzY8 (Part 3)**
>
> > **Q4.** In Figure 7 and 8, could the authors explain why increasing the number of layers in the branch net does not reduce the error of DeepONet. For the HyperDeepONet, why it may even increase the error? What is the intuition behind this phenomenon?
>
> **A.** We combined Figure 8 and Figure 9 into Figure 11 in the revised manuscript. We also added a training error graph for better understanding. Figure 11 compares the relative L2 error of the training data and test data for the DeepONet and the HyperDeepONet by varying the number of layers in the branch net and hypernetwork while maintaining the same small target network. The training and test error for the DeepONet is not reduced despite the depth of the branch net becoming larger. This is a limitation of DeepONet’s linear approximation. DeepONet approximates the operator with the dot product of the trunk net's output that approximates the basis of the target function and the branch net’s output that approximates the target function’s coefficient. Even if a more accurate coefficient is predicted by increasing the depth of the branch net, the error does not decrease because there is a limit to approximating the operator with a linear approximation using the already fixed trunk net.
>
> The HyperDeepONet approximates the operator with a low test error in all cases with a different number of layers. Figure 11 shows that the training error of the HyperDeepONet remains small as the depth of the hypernetwork increases, while the test error increases. The increasing gap between the training and test errors is because of overfitting. HyperDeepONet overfits the training data because the learnable parameters of the model are more than necessary to approximate the target operator.
>
> > **Q5.** When considering the whole complexity (i.e., the same total Params in table 2), the performance of Hyper seems to be inferior to that of DeepONet. How about the performance of other baselines?
>
> **A.** For three different PDEs with complicated target functions, we compare all the baseline methods in Table 7 to evaluate the performances. We analyze the model’s computation efficiency based on the number of parameters and fix the model’s complexity for each equation. All five models demonstrated their prediction abilities for the advection equation. DeepONet shows the greatest performance in this case, and other variants can no longer improve the performance. For the Burgers’ equation, NOMAD and HyperDeepONet are the two outstanding algorithms from the perspective of relative test error. NOMAD seems slightly dominant to our architectures, but the two models compete within the margin of error. Furthermore, HyperDeepONet improves its accuracy using the chunk embedding method, which enlarges the target network’s size while maintaining the complexity. Finally, HyperDeepONet and NOMAD outperform the other models for 2-dimensional shallow water equations. The HyperDeepONet still succeeds in accurate prediction even with few parameters. It can be observed from Table 7 that NOMAD is slightly more sensitive to an extreme case.
>
> > **Q6.** The authors mention that real-time prediction on resource-constrained hardware is crucial and challenging, so for each method, how about the training time, and inference time to obtain a desired prediction?
>
> **A.** Table 8 shows the training time and the inference time for the DeepONet and the HyperDeepONet for two different operator problems. When the same small target network is employed for the DeepONet and the HyperDeepONet, the training time and inference time for the HyperDeepONet are larger than for the DeepONet. However, in this case, the time is meaningless because DeepONet does not learn the operator with the desired accuracy at all (Table 1 and Figure 6).
>
> Even when both models use the same number of training parameters, HyperDeepONet takes slightly longer to train for one epoch than the DeepONet. However, the training complex operator using the HyperDeepONet takes fewer epochs to get the desired accuracy than DeepONet, as seen in Figure 7. This phenomenon can also be observed for the shallow water problem in Figure 12. It shows that the HyperDeepONet converges to the desired accuracy faster than any other variants of DeepONet.
>
> The HyperDeepONet also requires a larger inference time because it can infer the target network after the hypernetwork is used to generate the target network’s parameters. However, when the input function’s sensor values are already fixed, the inference time to predict the output of the target function for various query points is faster than that of the DeepONet. This is because the size of the target network for HyperDeepONet is smaller than that of the DeepONet, although the total number of parameters is the same.

---

> ### Author Response · Authors · 2022-11-18
> **Replies to Reviewer LzY8 (Part 4)**
>
> > **Q7.** The writing of this paper could be further polished. Many typos also exist. The writing is not good. I believe this work could be further improved with better writing
>
> **A.** We agree on this point. There were some awkward and unclear sentences, including the typos that you pointed out. We read the whole paper and revised the sentences more clearly. You can see it in the updated manuscript.
>
>
> *[1] Li, Zongyi, et al. "Fourier neural operator for parametric partial differential equations." _arXiv preprint arXiv:2010.08895_ (2020).*
>
> *[2] Takamoto, Makoto, et al. "PDEBENCH: An Extensive Benchmark for Scientific Machine Learning." _arXiv preprint arXiv:2210.07182_ (2022).*
>
> *[3] Lu, Lu, et al. "A comprehensive and fair comparison of two neural operators (with practical extensions) based on fair data." _Computer Methods in Applied Mechanics and Engineering_ 393 (2022): 114778.*
>
> *[4] Von Oswald, Johannes, et al. "Continual learning with hypernetworks." _arXiv preprint arXiv:1906.00695_ (2019).*

---

> ### Comment · Reviewer_LzY8 · 2022-11-19
> **Response to the rebuttal**
>
> I just had a quick view of all responses. I appreciate the authors for providing detailed explanations and extra experimental evaluation. Most of my concerns have been addressed well.
>
> **I would like to have a quick discussion with the authors here:**
> * In the answer to Q6:
>
> > a."When the same small target network is employed for the DeepONet and the HyperDeepONet"
> >
> > b. "Even when both models use the same number of training parameters"
>
> For setting a, what is the threshold (i.e., the upper bound of num of parameters) to decide whether a target network is small or not? Is there a consensus in the community? Also, how do you choose the num of parameters for setting b?
> My concern is whether the empirical observations still hold if the chosen number of parameters is different.
>
> * Minor: Please carefully check your citation format in the final version, e.g., [1], [2] and [4] are accepted by ICLR 2021, NeurIPS 2022 and ICLR 2020, respectively.

---

> > ### Author Response · Authors · 2022-11-20
> > **Thank you for your comments**
> >
> > We appreciate that the reviewer presents possible confusing points in our experimental settings. For the other baseline algorithms, we chose the settings for (b) including the structure of the models in each paper which presented DeepONet and its variants. For example, the authors in [3] offer the structures of DeepONet for the Burgers’ equation and the advection equation. We construct the HyperDeepONet architecture with the same complexity as other existing models.
> >
> > Based on the setting of the models for (b), we select the number of parameters for a small target network in (a). We observed that the dramatic differences in the model performance occur by reducing the complexity gradually. We set the threshold of the small target network for (a) when this difference occurs. Since the word ‘small target network‘ may be ambiguous and could be interpreted in many different ways, we include extra experiments in Appendix F, which consider the additional structures with different complexity.  Therefore, the empirical observations still hold for the different number of parameters although our model will also lose accuracy if the number of parameters is too small.
> >
> > We also appreciate the reviewer for pointing out the incorrect citations in our paper. We have revised the manuscript with careful attention to the publication of each paper. You can see it in the updated manuscript.

---

> > > ### Comment · Reviewer_LzY8 · 2022-11-20
> > > **Author rebuttal acknowledgement**
> > >
> > > Thanks for the clarification. The authors have addressed my concerns and updated the draft accordingly. Therefore, I will raise my initial score and recommend this work.

---

### Author Response · Authors · 2022-11-18
**General response to all Reviewers**

We would like to thank you all for your valuable feedback and suggestions. According to your comments, we have made many improvements to our paper and have uploaded the revised manuscript.

We have made the following changes to the manuscript

* We check and revise the manuscript for language, readability, clarity, and an appropriate tone.
* We add an experiment for more practical problem in Section 5-solution operator of the shallow water equation.
* We add an experiment on extending the HyperDeepONet using the chunk embedding method (c-HyperDeepONet) for the scalability of the HyperDeepONet in Section 5.
* We add the explanation of DeepONet in Appendix B.
* We add additional experiments to demonstrate the contribution of HyperDeepONet in Appendix F

All essential changes in the new version are highlighted in red color.  All the suggestions raised by the reviewers have been incorporated in the revised version of the manuscript.

Especially, the reviewers commonly pointed out that the writing needs improvements since many sentences are unclear with many typos. We agree on this point and have tried to put the details you pointed out in the text. Furthermore, according to your comments, we have improved the readability for the section containing the theory so that readers can follow the section easily. Here's a more intuitive and easy-to-understand explanation of this theory to help you understand.

In this study, we propose HyperDeepONet, an extension of the deep operator network (DeepONet) to overcome the limitation of the DeepONet even with limited resources. Section 4.2 provides the theoretical evidence to explain that the HyperDeepONet entails a relatively lower complexity than the DeepONet. The target network of HyperDeepONet corresponds to the trunk net of DeepONet. Theorem 2, our main theorem,  implies that the complexity of the trunk net inevitably grows fast to achieve a small error. The theorems in Appendix C state that analogous results hold for variants of DeepONet. We remark that the last theorem considers the overall complexity including the branch net, which corresponds to the hypernetwork of our model.

We briefly summarized the implications of theorems at the front of each paragraph in Section 4.2.  The core of proofs is based on Theorem 1 which offers a guideline on the neural network architecture for operator learning. It suggests that if the entire architecture can be replaced with a fully connected neural network, large complexity should be required for proper performance. We can substitute a neural network for the inner product between the outputs of the branch net and the trunk net, so that the DeepONet exactly faces the above issue. The theorems in Appendix C can be derived by observing that a small neural network can replace nonlinear reconstructors used in the variants of DeepONet.

We address other comments, questions, and concerns from reviewers in individual responses. We are very happy to communicate with you during the discussion period.

---

### Decision · Program_Chairs · 2023-01-20

**Decision:**

Accept: poster

**Justification For Why Not Higher Score:**

Good paper with strong scores. This is however focused on a  specific model with a limited impact.

**Justification For Why Not Lower Score:**

Good evaluation by all the reviewers.

**Metareview: Summary, Strengths And Weaknesses:**

The paper presents and extension of DeepONet a popular and recent neural operator introduced for solving PDEs and for modeling physical dynamics. The extension consists in modulating the weights of the predictor used in DeepONet, through a hypernetwork conditioned on a sample from an input function representing the system state at some time. The main contribution consists in analyzing the complexity of this family of networks, showing that the proposed approach is more efficient than previous ones. Experiments are performed on some classical benchmarks.

The strength of the paper lies in the theoretical analysis of the complexity of the DeepONet family which allows the authors to compare different variants of the model that can be described as particular cases of their hypernetwork based version. The algorithmic novelty itself is more limited: the proposed model makes use of a classical strategy consisting in conditioning the predictor parameters on an input function via a hypernetwork. Overall the approach is well motivated and obtains better performance than previous ones on the experimental evaluation. The authors answered to the reviewers’ comments by making significant additions to the manuscript and by adding experiments on a new dataset.  The manuscript should be proofread before the final submission.


**Note From Pc:**

if the above contains the word "oral" or "spotlight" please see: "oral" presentation means -> notable-top-5% and "spotlight" means -> notable-top-25%. As stated in our emails, we are disassociating presentation type from AC recommendations